# BIK1 protein homeostasis is maintained by the interplay of different ubiquitin ligases in immune signaling

Jiaojiao Bai [1,2,3,4,5], Yuanyuan Zhou [1,2,4,5], Jianhang Sun [1,4,5], Kexin Chen [1,4], Yufang Han [1,4], Ranran Wang [1,4], Yanmin Zou [1], Mingshuo Du [1,4] & Dongping Lu [2] ✉

Pathogen-associated molecular patterns (PAMPs) trigger plant innate immunity that acts as the first line of inducible defense against pathogen infection. A receptor-like cytoplasmic kinase BOTRYTIS-INDUCED KINASE 1 (BIK1) functions as a signaling hub immediately downstream of multiple pattern recognition receptors (PRRs). It is known that PLANT U-BOX PROTEIN 25 (PUB25) and PUB26 ubiquitinate BIK1 and mediate BIK1 degradation. However, how BIK1 homeostasis is maintained is not fully understood. Here, we show that two closely related ubiquitin ligases, RING DOMAIN LIGASE 1 (RGLG1) and RGLG2, preferentially associate with the hypo-phosphorylated BIK1 and promote the association of BIK1 with the co-receptor for several PRRs, BRI1-ASSOCIATED RECEPTOR KINASE1 (BAK1). PUB25 interacts with RGLG2 and mediates its degradation. In turn, RGLG2 represses the ubiquitin ligase activity of PUB25. RGLG1/2 suppress PUB25-mediated BIK1 degradation, promote BIK1 protein accumulation, and positively regulate immune signaling in a ubiquitin ligase activity-dependent manner. Our work reveals how BIK1 homeostasis is maintained by the interplay of different ubiquitin ligases.

Plants rely on innate immunity to defend against pathogens, which is triggered upon perception of microbe- or pathogen-associated molecular patterns (MAMPs/PAMPs) and bacterial effectors[1]. So far, an array of pattern recognition receptors (PRRs) perceiving corresponding PAMPs have been identified. For example, FLAGELLIN-SENSING2 (FLS2) recognizes the bacterial flagellin (or its derived peptide flg22)[2], and EF-TU RECEPTOR (EFR) perceives the bacterial elongation factor EF-Tu (or its derived epitope elf18)[3]. BRI1-ASSOCIATED RECEPTOR KINASE1 (BAK1) acts as the co-receptor for FLS2, EFR, and some other PRRs[4,5].

BOTRYTIS-INDUCED KINASE 1 (BIK1) and other closely related receptor-like cytoplasmic kinases (RLCKs) work as a signaling hub immediately downstream of multiple PRRs[6,7]. Upon PAMP perception,

BIK1 undergoes hyper-phosphorylation and directly phosphorylates the NADPH oxidase RESPIRATORY BURST OXIDASE HOMOLOG D (RbohD)[8,9], the $Ca^{2+}$-permeable channel OSCA1.3[10], and two CYCLIC NUCLEOTIDE-GATED CHANNEL (CNGC) proteins, CNGC2 and CNGC4[11], resulting in reactive oxygen species (ROS) burst, calcium entry into the cytosol, and stomatal immunity.

Both the kinase activity and protein stability of BIK1 are tightly controlled through the interplay between phosphorylation and ubiquitination[12]. BIK1 undergoes ligand-induced monoubiquitination that is mediated by the transmembrane ubiquitin ligases RING-H2 FINGER A3A (RHA3A) and RHA3B[13]. Such monoubiquitination of BIK1 plays a critical role in BIK1 endocytosis, release of BIK1 from the FLS2 complex, and immune signaling activation[13].

[1]State Key Laboratory of Plant Genomics, Center for Agricultural Resources Research, Institute of Genetics and Developmental Biology, Chinese Academy of Sciences, Shijiazhuang, Hebei 050021, China. [2]Joint Center for Single Cell Biology, School of Agriculture and Biology, Shanghai Jiao Tong University, Shanghai 200240, China. [3]College of Pharmacy and Life Science, Jiujiang University, Jiujiang, Jiangxi 332000, China. [4]University of Chinese Academy of Sciences, Beijing 100049, China. [5]These authors contributed equally: Jiaojiao Bai, Yuanyuan Zhou, Jianhang Sun. ✉e-mail: dplu@sjziam.ac.cn

Two ubiquitin ligases PLANT U-BOX PROTEIN 25 (PUB25) and PUB26 selectively polyubiquitinate the hypo-phosphorylated BIK1 and target it for proteasomal degradation. While the phospho-mimetic variant of BIK1, BIK1S236D/T237D, is resistant to ubiquitination by PUB25[14]. Furthermore, PUB25 and PUB26 are activated by phosphorylation on residues located in the region between the U-box and ARM domains, which is catalyzed by a calcium-dependent protein kinase 28 (CPK28)[14,15]. Intriguingly, we recently reported that CPK28 undergoes ubiquitination and proteasomal degradation that are induced by flg22 treatment. Two closely related transmembrane ubiquitin ligases ARABIDOPSIS TÓXICOS EN LEVADURA 31 (ATL31) and ATL6 interact with and polyubiquitinate CPK28, resulting in its degradation[16,17]. Moreover, PUB2 and PUB4 interact with BIK1, and PUB4 enhances BIK1-FLS2 and BIK1-RBOHD associations[18]. Interestingly, PUB4 was also shown to have a dual effect on BIK1 homeostasis: it mediates the ubiquitination and degradation of non-activated BIK1 at the resting state, but promotes the accumulation of activated BIK1 after PAMP treatment[19].

Although the regulatory mechanisms of BIK1 turnover have been studied, how BIK1 homeostasis is maintained is not fully understood. In this work, we found that two closely related RING-type ubiquitin ligases, RING DOMAIN LIGASE 1 (RGLG1) and RGLG2 preferentially interact with the hypo-phosphorylated BIK1 and directly ubiquitinate BIK1. Moreover, RGLG2 represses the ubiquitin ligase activity of PUB25, and PUB25 in turn mediates the proteasomal degradation of RGLG2. Together, the interplay of RGLG1/2 and PUB25 maintains BIK1 protein homeostasis and regulates immune signaling in plants.

## Results

### Identification of ubiquitin ligases that associate with BIK1

BIK1 functions as a signaling hub of plant immunity, its activity, subcellular localization, and protein stability are tightly regulated[12]. To identify potential ubiquitin ligases that regulate BIK1 homeostasis, we first screened ubiquitin ligase genes that share the similar expression profile with *BIK1*. We performed RNA sequencing (RNA-seq) to analyze the dynamic transcriptional responses at different time points (0, 15, 30, 60, 120, 240, 360, 480 min) after flg22 treatment (Supplementary Data 1). The differentially expressed genes (DEGs) were identified by comparison with time 0 [$\log_2$(fold change, FC) > 1, $P_{adj}$ < 0.05]. The number of DEGs rapidly increased from 15 to 60 min after flg22 treatment (Supplementary Fig. 1a). The expression of a number of DEGs was confirmed and analyzed by RT-qPCR (Supplementary Fig. 1b).

To group DEGs after flg22 treatment, the *K*-means clustering was employed and the optimal number of clusters (*K*) was determined based on the Akaike information criterion (AIC). The differentially expressed genes were grouped into 13 clusters (cluster 0–12) using the STEM (Short Time-series Expression Miner version 1.3.13) software, which was used to analyze time-course gene expression data[20] (Fig. 1a); after *K* = 13, the AIC keeps relatively stable (Supplementary Fig. 1c).

Four primary expression patterns (Pattern #1–4) were obtained and *BIK1* was in the cluster 10 with the expression pattern #2 (increasing rapidly then decreasing) (Fig. 1a, b). Notably, many well-known immune related genes also exhibited the #2 expression pattern, such as *BAK1, PEPR1/2, MPK3/4, CPK5, EDS1, RbohD, BSK1, RIN4, RPS4/5, ZAR1, BIR1, XLG2, RHA3B, PUB23/24, ATL6*, etc. (Fig. 1c). Then the DEGs with the expression pattern #2 were subjected to GO functional analysis, and they are mainly enriched in innate immune response, defense response to bacterium, ubiquitin-dependent protein catabolic process, response to stress, and endosome (Fig. 1d). Therefore, we suspect that the DEG clusters with the expression pattern #2 likely have other unidentified genes involved in immune regulation, including E3 ubiquitin ligase genes regulating BIK1 homeostasis. There are 179 ubiquitin ligase genes in these clusters (Supplementary Data 2). As BIK1 is known to be associated with the

plasma membrane (PM)[21], we focused on the 16 ubiquitin ligases that were predicted to be localized to the PM (http://csspalm.biocuckoo.org, https://dtu.biolib.com/DeepTMHMM) (Supplementary Fig. 2a).

To identify ubiquitin ligases that could associate with BIK1, we performed split-luciferase complementation (SLC) assays, in which BIK1-CLuc (CL) and E3-NLuc (NL) were expressed in *Nicotiana benthamiana*. PUB25-NLuc was used as the positive control of the BIK1 interactor. Of the 16 ubiquitin ligases tested, 6 ones (RGLG1, ATL83, ATL32, ATL6, ATL40, and ATL80) were found to be associated with BIK1 (Supplementary Fig. 2b). Furthermore, the association of BIK1 with RGLG1, ATL83, ATL32, or ATL6 was reduced upon flg22 stimulation as assayed by SLC in *N. benthamiana* (Supplementary Fig. 2c). We also confirmed the association of BIK1 with ATL6 by co-immunoprecipitation (IP) assays in Arabidopsis protoplasts. Notably, flg22 treatment reduced their association (Supplementary Fig. 3a). The association of BIK1 with ATL40 or ATL80 was almost not affected by flg22 treatment as assayed by SLC (Supplementary Fig. 2c), and that with ATL80 was confirmed by co-IP assays in Arabidopsis protoplasts (Supplementary Fig. 3b). We reported recently that ATL6/31 associated with CPK28, a negative immune regulator that was shown to directly interact with BIK1[16,17]. Therefore, ATL6 and BIK1 are very likely existing in the same protein complex. We next focused on RGLG1 in this work and examined whether RGLG1 plays roles in regulating BIK1 homeostasis.

### RGLG1/2 prefer to associate with the hypo-phosphorylated BIK1

RGLG2 is the closest homolog of RGLG1, and they belong to a five-member RGLG family (RGLG1-5) (Supplementary Fig. 4a–b)[22]. Both RGLG1-GFP and RGLG2-GFP were localized to the PM, when they were expressed in Arabidopsis protoplasts (Supplementary Fig. 4c–d). RGLG1/2 were shown to have ubiquitin ligase activity[23]. The C425/C428/C454/C457 sites chelating $Zn^{2+}$,[22] in the RING domain of RGLG2 are conserved in AtRGLGs and RGLG1/2 homologs from rice (*Oryza sativa*), maize (*Zea mays*), and *Brachypodium distachyon* (Supplementary Fig. 5a). Mutation of these sites in RGLG2 (RGLG2m) and the corresponding sites in RGLG1 (RGLG1m) largely impaired their ubiquitin ligase activity (Supplementary Fig. 4a, 5a–c), as assayed in a bacterial ubiquitination system that was previously developed in our laboratory[24], where AtUBA1, AtUBC8, RGLG1/2-MYC, or RGLG1m/2m-MYC as well as FLAG-tagged ubiquitin (His-FLAG-Ub) were co-expressed in *Escherichia coli* BL21 strain (Supplementary Fig. 5b–c).

We found that, like RGLG1-NLuc (Supplementary Fig. 2b, c), RGLG2-NLuc also associated with BIK1-CLuc as demonstrated by SLC assays in *N. benthamiana* (Fig. 2a). Similar results were also obtained when BIK1-NLuc and RGLG1/2-CLuc were transiently expressed in *N. benthamiana* (Supplementary Fig. 6a–b). Interestingly, treatment of *N. benthamiana* with flg22 largely reduced the association between BIK1 and RGLG1/2 (Fig. 2a, Supplementary Fig. 2c), suggesting that RGLG1/2 might dissociate from BIK1 upon ligand perception.

To confirm the association of RGLG1/2 with BIK1, we co-expressed FLAG-epitope-tagged RGLG1/2 and HA-epitope-tagged BIK1 in Arabidopsis protoplasts. RGLG1/2-FLAG proteins were immunoprecipitated by anti-FLAG antibodies, and BIK1-HA was present in the RGLG1/2-FLAG immunoprecipitates (Fig. 2b, c). Similarly, RGLG1/2-HA were also present in the BIK1-FLAG immunoprecipitates (Fig. 2d, Supplementary Fig. 6c). Importantly, the association between BIK1 and RGLG1/2 was reduced when the protoplasts were treated with flg22 (Fig. 2b–d and Supplementary Fig. 6c). Co-IP assays using the anti-RGLG1 antibodies also showed that the association of RGLG1 with BIK1 was reduced when the protoplasts were exposed to flg22 (Supplementary Fig. 6d–e). Whereas, RGLG3 did not associate with BIK1 (Supplementary Fig. 7a). Moreover, RGLG2 did not associate with FLS2 under the same conditions (Supplementary Fig. 7b). Neither RGLG1 nor RGLG2 is able to associate with another RLCK, AvrPphB SUSCEPTIBLE1-LIKE13 (PBL13)

that is a negative immune regulator that phosphorylates RbohD to affect its activity and stability[25,26] (Supplementary Fig. 7c–d).

We also performed co-IP assays using stable transgenic lines. RGLG2 was associated with BIK1 in *pRGLG2::RGLG2-HA/p35S::BIK1-FLAG* plants (Fig. 2e), and RGLG1 was associated with BIK1 in *p35S::RGLG1-HA/p35S::BIK1-FLAG* plants (Supplementary Fig. 7e). Consistently, the association of RGLG1/2 with BIK1 was reduced upon flg22 treatment (Fig. 2e and Supplementary Fig. 7e).

We also performed bimolecular fluorescence complementation (BiFC) assays in Arabidopsis protoplasts using BIK1 tagged with an nYFP (the N-terminal half of yellow fluorescent protein) tag and RGLG1/2 tagged with a cYFP (the C-terminal half of YFP) tag. BIK1 associated with RGLG1/2 at the PM. Whereas, BIK1-nYFP did not associate with RGLG3-cYFP, and RGLG1/2-cYFP did not associate with PBL13-nYFP (Fig. 2f and Supplementary Fig. 7f). These results verify the specificity of the association between BIK1 and RGLG1/2 at the PM.

Furthermore, both RGLG1 and RGLG2 were pulled down by BIK1 fused to a GST tag, but not by GST or GST-FLS2CD (the intracellular region of FLS2) in in vitro assays (Fig. 2g, Supplementary Fig. 8a). These results suggest that RGLG1/2 directly interact with BIK1. RGLG2 proteins contain a so-called copine (C, or von Willebrand factor type A) domain and a RING (R) domain[23] (Supplementary Fig. 8b), and both the C domain and the R domain are conserved in AtRGLGs and RGLG homologs from rice, maize, and *B. distachyon* (Supplementary Fig. 5a). The region at the N-terminus is called S domain, and the region lying between the Copine and the RING domains is called D domain[23] (Supplementary Fig. 8b). We found that BIK1 directly interacts with the RGLG2 variant (RGLG2SC) lacking both the R domain and the D domain (Supplementary Fig. 8c), suggesting that the S and C domains are likely responsible for the association of RGLG2 with BIK1. However, either the S domain or the C domain alone did not interact with BIK1 (Supplementary Fig. 8c–d). These results indicate that the S-C domains of

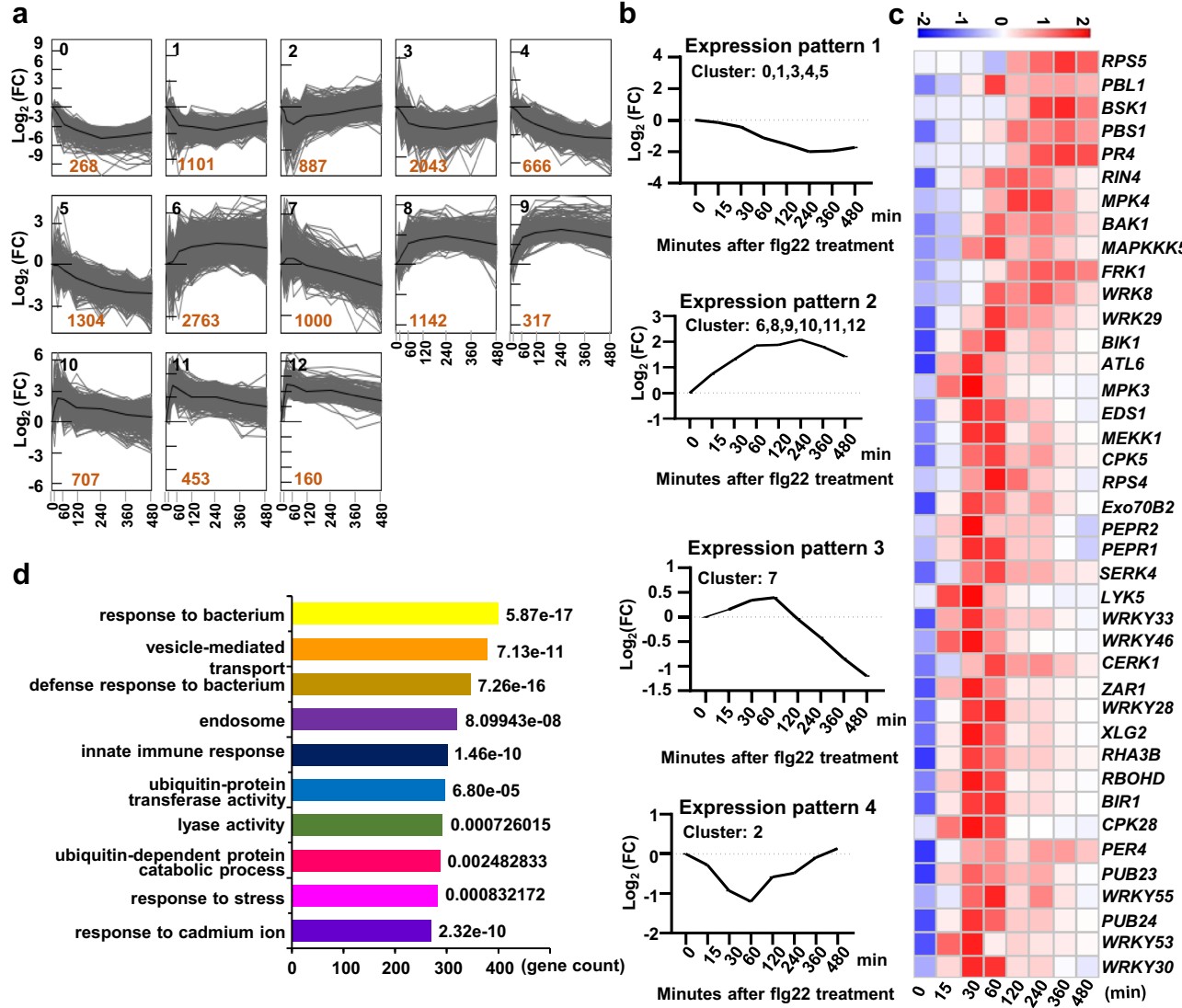

**Fig. 1 | The dynamic gene expression upon persistent flg22 treatment as analyzed by RNA-seq. a** The deferentially expressed genes (DEGs) were grouped into 13 clusters. Twelve-day-old Arabidopsis seedlings were treated with 5 μM flg22 for the indicated times. Totally 13409 DEGs (log₂[fold change, FC] > 1, $P_{adj}$ < 0.05) were grouped via *K*-means clustering by STEM software. The black number indicates the cluster ID, and the brown number shows the number of genes in each cluster. The gray lines show the dynamic gene expression profile during the 8 h time course of flg22 treatment, and the centroid gene expression profile was plotted by the dark black lines. The FPKM value for each time point was the mean of three biological replicates. **b** Four gene expression patterns (#1–4) during the 8 h time course of flg22 treatment. **c** The expression heatmap of immune related genes with the expression pattern #2, based on the FPKM values of three biological replicates. **d** GO enrichment analysis of DEGs with the expression pattern #2, GO terms were determined using the agriGo database. The top 10 biology process (BP) terms were shown.

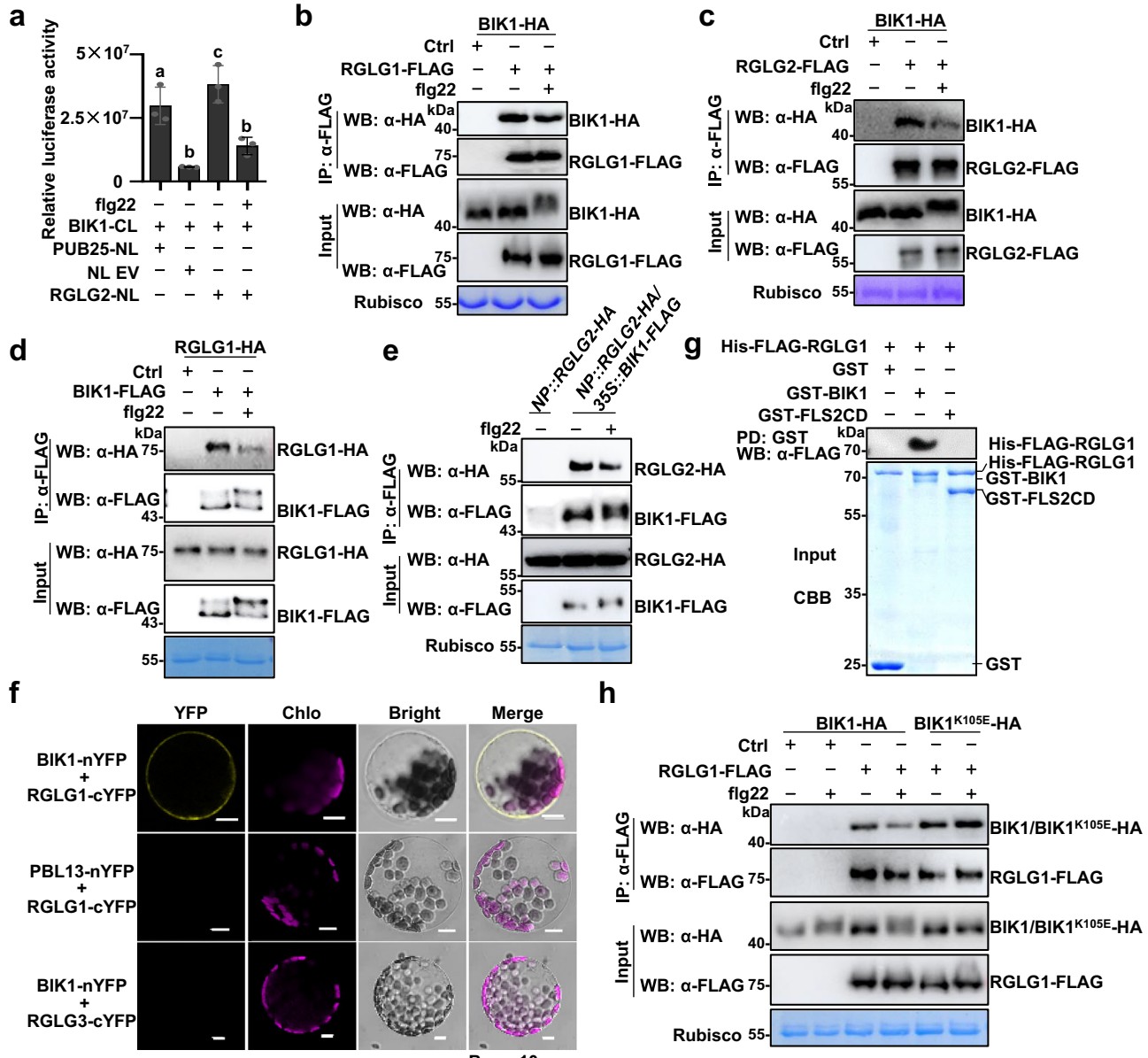

**Fig. 2 | RGLG1/2 preferentially associate with the hypo-phosphorylated BIK1.**
**a** BIK1 associates with RGLG2 as assayed by split-luciferase complementation. BIK1-CLuc (CL) and RGLG2-NLuc (NL) were expressed in *Nicotiana benthamiana*. The *N. benthamiana* leaves were treated with or without 5 μM flg22 for 30 min before harvesting. PUB25-NLuc and empty NLuc vector (EV) were used as positive and negative controls, respectively. Relative luciferase activity was quantified and the values are means ± SD ($n$ = 3 leaves from three independent biological repeats). Different letters denote significance difference (one-way ANOVA, $P$ = 0.0003). Source data are provided as a Source Data file. **b, c** RGLG1/2 associate with BIK1 as assayed by co-IP. RGLG1/2-FLAG and BIK1-HA were co-expressed in Arabidopsis protoplasts. The protoplasts were treated with or without 2.5 μM flg22 for 30 min. RGLG1/2-FLAG was immunoprecipitated with anti-FLAG antibodies, the associated BIK1-HA proteins were detected by immunoblotting with anti-HA antibodies. **d** RGLG1 associate with BIK1 as assayed by co-IP. RGLG1-HA and BIK1-FLAG were co-expressed in Arabidopsis protoplasts. The protoplasts were treated with or without 2.5 μM flg22 for 30 min. BIK1-FLAG was immunoprecipitated with anti-FLAG

antibodies. **e** RGLG2 is associated with BIK1 in stable transgenic plants. *pRGLG2* (*NP*)::*RGLG2-HA/p35S::BIK1-FLAG* and *pRGLG2*(*NP*)::*RGLG2-HA*/Col-0 seedlings were treated with 5 μM flg22 for 30 min. BIK1-FLAG was immunoprecipitated with anti-FLAG antibodies. **f** RGLG1 associates with BIK1 at the plasma membrane. The indicated BiFC constructs were transfected into Arabidopsis protoplasts, and fluorescence was visualized by confocal microscopy. The scale bar = 10 μm. **g** RGLG1 directly interacts with BIK1 but not FLS2CD. The recombinant GST-BIK1 or GST-FLS2CD proteins immobilized on GSH beads were incubated with His-FLAG-RGLG1 proteins. GST pull-down was performed and the pulled-down His-FLAG-RGLG1 proteins were detected by immunoblotting with anti-FLAG antibodies. All input proteins were separated by SDS-PAGE and stained with Coomassie Brilliant Blue (CBB). **h** Flg22 treatment does not affect the association between RGLG1 and BIK1[K105E]. RGLG1-FLAG and BIK1-HA or BIK1[K105E]-HA were co-expressed in protoplasts. The protoplasts were treated with flg22 and co-IP was performed as described in (**b**). **b**–**h** Images shown are representative of at least two independent experiments.

RGLG2 work as a module to interact with BIK1, which may be responsible for the substrate recognition.

It is known that flg22 induces BIK1 phosphorylation, which is evident by a mobility shift in SDS-PAGE[6,7]. Interestingly, we found that the association of RGLG1/2 with the BIK1 kinase dead mutant, BIK1[K105E]

(BIK1Km)[6] was stronger than with wild-type (WT) BIK1. Moreover, the flg22-induced dissociation was not observed between RGLG1/2 and BIK1Km (Fig. 2h and Supplementary Fig. 9a). By contrast, the association of RGLG1/2 with the BIK1 phospho-mimetic mutant BIK1S236D/T237D (BIK1[2D])[7,14] was weaker than with WT BIK1. Notably, the association of

RGLG1/2 with BIK1[2D] was further reduced upon flg22 treatment (Supplementary Fig 9b–c), suggesting that flg22 induces phosphorylation at other sites besides S236 and T237[27]. As BIK1Km did not undergo flg22-induced phosphorylation[28] (Fig. 2h), these results suggest that RGLG1/2 preferentially associate with the hypo-phosphorylated BIK1.

## RGLG1/2 positively regulate immune signaling

To investigate the role of RGLG1/2 in regulating immunity, we generated the *rglg1 rglg2* double mutant (Supplementary Fig. 10a–c). Notably, unlike those in *bik1*, the *PR1* transcript levels were slightly but not significantly higher in *rglg1 rglg2* than in Col-0 (Supplementary Fig. 10d). BIK1 is essential for reactive oxygen species (ROS) burst after flg22 treatment[8,9]. We therefore tested whether RGLG1/2 regulate the flg22-induced ROS burst. After flg22 treatment, the *rglg1* and *rglg2* single mutants exhibited slightly but not significantly less ROS production than Col-0 plants, and the *rglg1 rglg2* double mutant plants produced significantly less ROS than Col-0 (Fig. 3a, Supplementary Fig. 11), suggesting that RGLG1 and RGLG2 play redundant and positive roles in regulating PTI signaling. MITOGEN-ACTIVATED PROTEIN KINASE (MAPK) activation is one of the early immune responses[29,30]. It is known that BIK1 is not involved in regulating MAPK activation[31,32]. We found that flg22-induced MPK3/6/4 activation was not affected by the loss of *RGLG1* and *RGLG2* (Fig. 3b).

To examine whether RGLG1/2 play a role in regulating disease resistance, we inoculated *rglg1 rglg2* and Col-0 plants with the bacterial pathogen *Pseudomonas syringae* pv. *tomato* (*Pst*) DC3000. We found that *rglg1 rglg2* plants supported significantly higher bacterial growth than Col-0 plants (Fig. 3c). Moreover, *rglg1 rglg2* were more susceptible than Col-0 to the nonvirulent type III secretion mutant strain *Pst* DC3000 *hrcC⁻* (Fig. 3d). These results demonstrate that RGLG1/2 positively regulate BIK1-mediated immunity.

## RGLG1/2 are required for BIK1 protein accumulation

It is known that PUB25 mediates the ubiquitination and subsequent degradation of BIK1[14]. We showed that another two E3 ubiquitin ligases RGLG1 and RGLG2 interact with BIK1 and positively regulate BIK1-mediated immunity. To examine whether RGLG1/2 affect BIK1 protein stability, we co-expressed BIK1 together with RGLG1/2 or PUB25 in Arabidopsis protoplasts. The protein synthesis inhibitor cycloheximide (CHX) is often used in the protein degradation assays to exclude the translational effect, so that the changes in protein levels should be the already translated proteins[14,19,33]. Consistent with the previous report[14], in the presence of CHX, PUB25 expression reduced the BIK1 protein abundance. By contrast, RGLG1/2 expression resulted in the increased accumulation of BIK1 in the presence of CHX (Fig. 3e, Supplementary Fig. 12a), suggesting that RGLG1/2 likely promote the protein accumulation of BIK1. However, the expression of RGLG1/2 reduced the protein abundance of ERF53 and PP2CA in the presence of CHX, both were shown to be the substrates of RGLG1 (Supplementary Fig. 12b, c)[34,35]. Notably, in the absence of CHX, overexpression of RGLG1 hardly affected the BIK1 protein accumulation in protoplasts (Supplementary Fig. 12d). This might be because transient protein overexpression in Arabidopsis protoplasts results in synthesis of large amounts of BIK1 proteins; therefore, the contribution of BIK1 degradation to its abundance becomes unconspicuous under this condition.

Then we generated *p35S::BIK1-FLAG* in the *rglg1 rglg2* double mutant background and detect BIK1 proteins in the transgenic plants. BIK1 protein levels were substantially lower in *p35S::BIK1-FLAG/rglg1 rglg2* than in *p35S::BIK1-FLAG/*Col-0 (Fig. 3f), while *BIK1* transcript levels in *p35S::BIK1-FLAG/rglg1 rglg2* were comparable to those in *p35S::BIK1-FLAG/*Col-0 (Supplementary Fig. 13a). The growth defect phenotype of *rglg1 rglg2* were almost not restored by overexpressing *BIK1* (Supplementary Fig. 13b). Accordingly, overexpressing *BIK1* in *rglg1 rglg2* only slightly but not significantly restores its immune phenotype as indicated by the bacteria growth in the transgenic plants (Supplementary

Fig. 13c). Moreover, the transcript levels of *PUB25* in *p35S::BIK1-FLAG/rglg1 rglg2* were comparable to those in *rglg1 rglg2* (Supplementary Fig. 13d); and the protein accumulation of exogenous PUB25-HA was not increased by *BIK1* overexpression (Supplementary Fig. 13e). These results suggest that BIK1 protein accumulation largely depends on RGLG1/2.

To further confirm this premise, we generated *pBIK1::BIK1-HA* (under the control of its native promoter) in the *rglg1 rglg2* double mutant background. To detect BIK1 proteins in *pBIK1::BIK1-HA/rglg1 rglg2*, we enriched the total proteins by isolating protoplasts from plenty of leaves from 4-week-old plants, and then expressed RGLG1/2/3 in *pBIK1::BIK1-HA/rglg1 rglg2* protoplasts. The protoplasts were treated with CHX for 2 h to exclude the translational effect. The results showed that both RGLG1 and RGLG2, but not RGLG3, promoted the accumulation of BIK1 proteins (Fig. 3g, Supplementary Fig. 14a). RHA3B was shown to mediate the monoubiquitination of BIK1 and the subsequent release of BIK1 from the FLS2 receptor complex[13]. However, RHA3B expression did not affect the BIK1 protein abundance in *pBIK1::BIK1-HA/rglg1 rglg2* protoplasts (Supplementary Fig. 14b).

Furthermore, we found that the expression of RGLG1/2, but not RHA3B, suppressed the PUB25-mediated BIK1 degradation (Fig. 3h–i and Supplementary Fig. 14c), implying that RGLG1/2 promote BIK1 accumulation likely through inhibiting BIK1 degradation that is mediated by PUB25/26.

Consistently, combined mutations of the C-terminal four lysines (BIK1[C4KR]) that are ubiquitination sites mediated by RHA3A/B[13] led to its inability to dissociate from FLS2 upon flg22 treatment (Supplementary Fig. 15a). However, the protein accumulation of BIK1[C4KR] promoted by RGLG1/2 was comparable to that of WT BIK1; and PUB25 mediated the degradation of BIK1[C4KR] as it did for WT BIK1 (Supplementary Fig. 15b–c). These results suggest that the role of RGLG1/2 in regulating BIK1 is distinct from that of RHA3A/B.

## The interplay between RGLG1/2 and PUB25

We next further explore the relationship between RGLG1/2 and PUB25. The results of co-IP assays showed that RGLG1/2 associated with PUB25 in Arabidopsis protoplasts. Furthermore, the association was reduced upon flg22 stimulation (Fig. 4a, b). We also performed BiFC assays, the results showed that PUB25-nYFP associated with RGLG1/2-cYFP at the PM, while PUB25-nYFP did not associate with RGLG3-cYFP, and RGLG1/2-cYFP did not associate with PUB39-nYFP (Fig. 4c and Supplementary Fig. 16a). Moreover, the results of pull-down assays showed that RGLG1/2 directly interacted with PUB25 in vitro (Fig. 4d and Supplementary Fig. 16b). Moreover, we found that PUB25 directly ubiquitinated RGLG2m in vitro (Supplementary Fig. 16c).

Interestingly, we found that, in the presence of CHX, the proteins levels of RGLG2 were reduced when they were co-expressed with PUB25, which was blocked by MG132, an inhibitor of 26S proteasome (Fig. 4e). Furthermore, when transiently expressed, RGLG1/2 protein levels were higher in *pub25 pub26* double mutant protoplasts than in Col-0 protoplasts in the presence of CHX (Fig. 4f, Supplementary Fig. 17a–b). These results suggest that PUB25 mediates the proteasomal degradation of RGLG1/2.

PUB25 has ubiquitin ligase activity as demonstrated by its autoubiquitination (Fig. 4g). Interestingly, expression of RGLG2, but not RGLG2m led to the reduction in PUB25 autoubiquitination (Fig. 4g). However, the non-covalent association of PUB25 with a Ub moiety was not affected (Fig. 4g). These results suggest that RGLG2 suppresses the ubiquitin ligase activity of PUB25.

## The ubiquitin ligase activity is required for the role of RGLG2 in regulating immunity

Unlike WT RGLG1/2, RGLG1m/2m hardly suppressed the PUB25-mediated BIK1 protein degradation (Fig. 5a, Supplementary Fig. 18a).

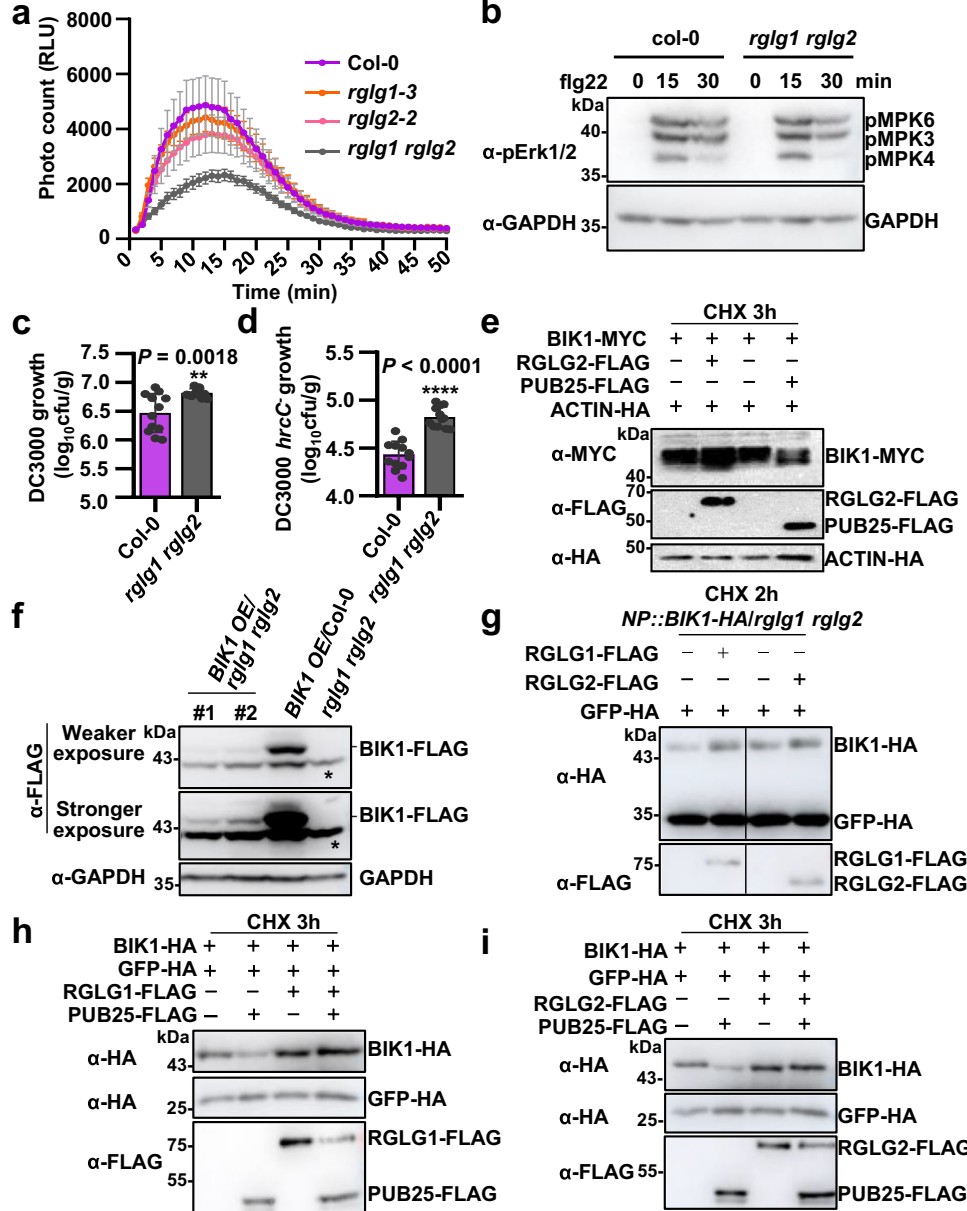

**Fig. 3 | RGLG1/2 promote BIK1 protein accumulation and positively regulate immunity. a** The *rglg1 rglg2* double mutant plants produce less flg22-induced ROS than Col-0. Leaf disks from five-week-old plants were treated with 200 nM flg22. ROS production represented as relative light unit (RLU) was immediately measured. The values were shown as means ± SEM ($n = 12$ leave discs, one-way ANOVA, $P = 0.0013$). Source data are provided as a Source Data file. **b** flg22-induced MPK3/6 activation in *rglg1 rglg2* and Col-0 plants. Seven-day-old seedlings were treated with 100 nM flg22 for the indicated times. MAPK activation was detected by immunoblotting with anti-pErk1/2 antibodies. GAPDH was detected as the loading control. Images shown are representative of three independent experiments. **c, d** *rglg1 rglg2* plants are more susceptible to bacterial pathogens. Thirty-day-old plants were infiltrated with *Pst* DC3000 or *Pst* DC3000 *hrcC⁻*. Bacterial growth was determined 3 days post inoculation and was evaluated as colony-forming units per gram of leaves (cfu/g). Individual data points were shown with means ± SD, for both (**c**) and (**d**), $n = 12$ leaves from four biological replicates using independent plants grown and inoculated with bacterial pathogens under the same conditions. Statistical significance compared with Col-0 was determined by Two-Tailed Student's *t*-tests: **$P = 0.0018$ (**c**), ****$P < 0.0001$ (**d**). Source **d**ata are provided as a Source Data file.

**e** RGLG2 promotes BIK1 protein accumulation. BIK1-MYC was co-expressed together with RGLG2-FLAG or PUB25-FLAG in protoplasts. The protoplasts were treated with 50 μM CHX for 3 h before harvesting. ACTIN-HA was used as an internal transfection control. The transiently expressed proteins were detected by immunoblotting with the corresponding antibodies. Images shown are representative of three independent experiments. **f** Detection of BIK1 proteins in *p35S::BIK1-FLAG* (*BIK1* OE)/*rglg1 rglg2* and *BIK1* OE/Col-0 transgenic plants. GAPDH was detected as the loading control. Asterisks indicate nonspecific bands. Images shown are representative of two independent experiments. **g** RGLG1/2 promote BIK1 accumulation. RGLG1/2-FLAG and GFP-HA were co-expressed in protoplasts isolated from *pBIK1(NP)::BIK1-HA/rglg1 rglg2* plants, The protoplasts were treated with 50 μM CHX for 2 h before harvesting. GFP-HA was used as an internal transfection control. Images shown are representative of three independent experiments. **h, i** RGLG1/2 suppress PUB25-mediated BIK1 degradation. BIK1-HA and GFP-HA were co-expressed together with RGLG1/2-FLAG or PUB25-FLAG in Arabidopsis protoplasts. The protoplasts were treated with 50 μM CHX for 3 h before harvesting. Images shown are representative of at least two independent experiments.

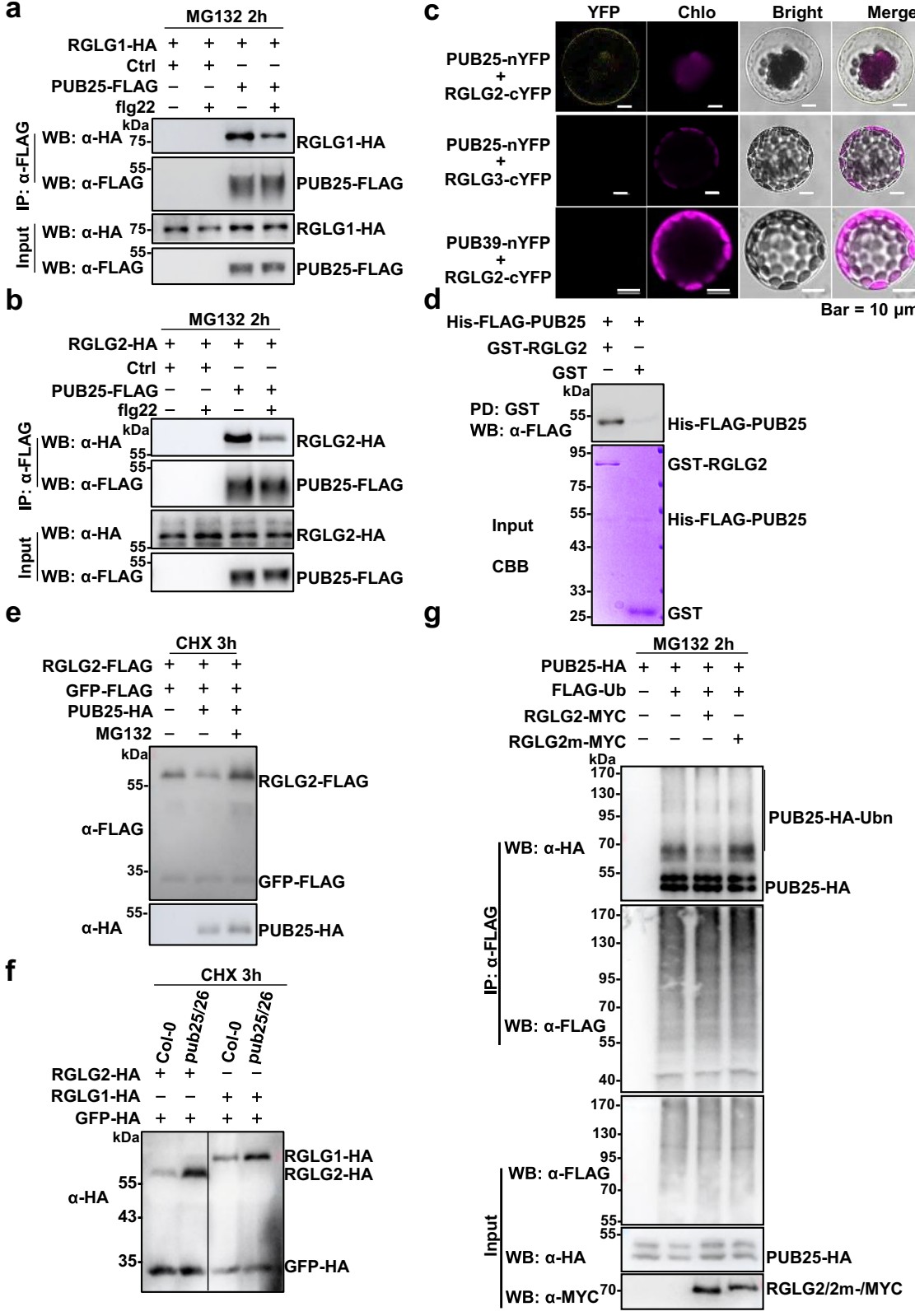

However, the association of BIK1 with RGLG1/2 was comparable to that with RGLG1m/2 m both in vivo and in vitro (Supplementary Fig. 18b–e). To confirm that the ubiquitin ligase activity of RGLG2 is required for its role in immunity, we generated transgenic lines expressing *RGLG2m* in the *rglg1 rglg2* background under its native promoter. WT *RGLG2* was able to complement both the growth and immune phenotypes of *rglg1 rglg2* (Fig. 5b–d). By contrast, *RGLG2m* only partially rescued the

growth-defective phenotype of *rglg1 rglg2* (Fig. 5b), slightly but not significantly restored the resistance of *rglg1 rglg2* to *Pst* DC3000 *hrcC⁻* (Fig. 5c), and almost did not restore the flg22-induced ROS production in *rglg1 rglg2* (Fig. 5d). The transcript levels of *RGLG2m* are comparable to those of WT *RGLG2* in these transgenic plants (Supplementary Fig. 18f). These results suggest that the ubiquitin ligase activity is required for the role of RGLG2 in regulating immunity.

**Fig. 4 | The interplay between RGLG1/2 and PUB25. a, b** PUB25 associates with RGLG1/2 as assayed by co-IP. RGLG1/2-HA and PUB25-FLAG were co-expressed in Arabidopsis protoplasts. The protoplasts were pretreated with 30 μM MG132 for 2 h, and treated with or without 2.5 μM flg22 for 30 min. PUB25-FLAG was immunoprecipitated with anti-FLAG antibodies. Images shown are representative of at least two independent experiments. **c** PUB25 associates with RGLG2 at the plasma membrane. The indicated BiFC constructs were transfected into Arabidopsis protoplasts, and fluorescence was visualized by confocal microscopy. Scale bars: 10 μm. **d** RGLG2 directly interacts with PUB25 in vitro. The recombinant GST-RGLG2/GST proteins immobilized on GSH beads were incubated with purified His-FLAG-PUB25 proteins. The pulled-down His-FLAG-PUB25 proteins were detected by immunoblotting with anti-FLAG antibodies. **c, d** images shown are representative of three independent experiments. **e** PUB25 mediates RGLG2 degradation. RGLG2-

FLAG was co-expressed with or without PUB25-HA in Arabidopsis protoplasts. GFP-FLAG was used as an internal transfection control. The protoplasts were treated with 50 μM CHX, and with or without 30 μM MG132 for 3 h before harvesting. **f** RGLG1/2 protein levels are higher in *pub25 pub26* than in Col-0 protoplasts. RGLG1/2-HA were expressed in *pub25 pub26* or Col-0 protoplasts. GFP-HA was used as an internal transfection control. The protoplasts were treated with 50 μM CHX for 3 h. **e, f** Images shown are representative of at least two independent experiments. **g** RGLG2 represses the ubiquitin ligase activity of PUB25. *PUB25-HA* and *FLAG-Ub* were co-transfected with *RGLG2-MYC* or *RGLG2m-MYC* into Arabidopsis protoplasts. The protoplasts were treated with 30 μM MG132 for 2 h before harvesting. IPs were performed with anti-FLAG antibodies. PUB25 autoubiquitination was detected by immunoblotting with anti-HA antibodies. Images shown are representative of three independent experiments.

RGLG1/2 directly ubiquitinated BIK1 in the bacterial system, where AtUBA1 (E1), AtUBC8 (E2), RGLG1/2-MYC (E3), His-FLAG-Ub, and MBP-BIK1-HA were co-expressed in *E. coli* BL21 strain (Fig. 5e, f). The lack of E1, E2, RGLG2, or ubiquitin resulted in the loss of BIK1 ubiquitination (Fig. 5e). As expected, ubiquitination of BIK1 by RGLG2m was significantly reduced (Supplementary Fig. 19a). Moreover, RGLG1/2 were not able to ubiquitinate PBL13 (Supplementary Fig. 19b–c). Although both PUB25 and RGLG1/2 ubiquitinate BIK1, the ubiquitination pattern mediated by RGLG1/2 was different from that by PUB25. The ubiquitination of BIK1 by PUB25 exhibited a laddering pattern, while that by RGLG1/2 seemed to be monoubiquitination (Fig. 5f).

We also observed that there was competition between RGLG1/2 and PUB25 in binding to BIK1. The results of co-IP showed that when RGLG2 was present, the association of BIK1 with PUB25 was reduced (Supplementary Fig. 20a). Moreover, either RGLG1 or RGLG2, but not RGLG3 reduced the interaction between BIK1 and PUB25 in vitro (Supplementary Fig. 20b). Notably, the co-IP results showed that association of BIK1 and PUB25 was also reduced by RGLG1m/2m expression, although to a less extent than by RGLG1/2 (Supplementary Fig. 20c–d), suggesting that RGLG1m/2m are also able to compete with PUB25 for BIK1, and the ubiquitin ligase activity of RGLG1/2 is partially required for their competition with PUB25. However, RGLG2m only slightly but not significantly restored the immune responses of *rglg1 rglg2* with respect to resistance to *Pst* DC3000 *hrcC*⁻ (Fig. 5c). These results imply that the competition of RGLG1/2 with PUB25 likely make a minor contribution to the function of RGLG1/2 in regulating BIK1 homeostasis.

## RGLG1/2 promote the BAK1-BIK1 interaction

RGLG2 was also associated with BAK1 in Arabidopsis protoplasts, and the association was not affected by the flg22 treatment (Fig. 6a). Moreover, BAK1 protein level was comparable in *rglg1 rglg2* and Col-0 plants (Supplementary Fig. 21a). We then tested whether the association of RGLG2 and BIK1 requires BAK1. RGLG2-BIK1 association was much weaker in the *bak1-4* mutant than in Col-0 plants (Fig. 6b), indicating that RGLG2-BIK1 association partially requires BAK1. Consistent with the previous report[6], BAK1-BIK1 association was reduced upon flg22 treatment (Supplementary Fig. 21b); consistently, RGLG2-BIK1 interaction was also reduced by the flg22 treatment (Supplementary Fig. 21c). These results suggest that RGLG2 is likely accompanying BAK1 in the same complex to regulate immunity.

RGLG2 enhanced the interaction between BIK1 and BAK1 both in vivo and in vitro (Fig. 6c, d). Furthermore, when BAK1-FLAG was co-expressed with BIK1-HA and RGLG1/2-HA in protoplasts, the co-IP results showed that both BIK1-HA and RGLG1/2-HA could be pulled down by BAK1-FLAG, and the results also confirmed that the association of BAK1 and BIK1 were promoted by RGLG1/2 (Fig. 6e, Supplementary Fig. 21d). Furthermore, the association of BAK1 and BIK1 was weaker in *rglg1 rglg2* than in Col-0 (Fig. 6f). These results suggest that BAK1, BIK1, and RGLG1/2 are in the same complex, and RGLG1/2

positively regulate immune signaling partially through promoting the interaction between BAK1 and BIK1.

## Discussion

BIK1 acts as an immune signaling hub that directly phosphorylates multiple downstream executors[8–11]. Both the kinase activity and protein stability of BIK1 are tightly regulated to ensure proper immune responses in plants[12]. So far, at least three pairs of ubiquitin ligases have been identified to associate with BIK1, which play distinct roles in regulating BIK1 turnover, subcellular localization, and interaction with its partners[13,14,18,19]. Here, we demonstrate that another two closely related RING-type ubiquitin ligases, RGLG1 and RGLG2 promote BIK1 accumulation and positively regulate BIK1-mediated immune signaling, and our work reveals that BIK1 protein homeostasis is maintained by interplay of RGLG1/2 and PUB25 (Fig. 7).

There are diverse types of ubiquitination, such as monoubiquitination and polyubiquitination. The polyubiquitination varies with diverse linkage types of Ub chains, such as K48-, K63-, and Met1-linkage[36]. The diverse outcomes of BIK1 ubiquitination mediated by different ubiquitin ligases further support the notion that the mode of ubiquitination dictates the fate of the ubiquitinated proteins. The K48-linked Ub chain always encodes a signal for 26 S proteasomal degradation of the modified proteins, while monoubiquitination or K63- and Met1-linked ubiquitination can also serve as a nonproteolytical signal, like regulating a substrate's activity, changing a substrate's localization, and recruiting other factors to modulate particular signaling pathways[36]. The type of BIK1 ubiquitination by PUB25/26 seems to be polyubiquitination[14] (Fig. 5f), while RHA3A/B mediate the monoubiquitination of BIK1[13]. In this work, we demonstrate that RGLG1/2 directly ubiquitinate BIK1, and the mode of BIK1 ubiquitination by RGLG1/2 seems to be monoubiquitination as demonstrated by in vitro assays (Fig. 5e, f and Supplementary Fig. 19). However, the context, mechanism, and outcome of BIK1 ubiquitination mediated by RHA3A/B are distinct from those by RGLG1/2. RGLG1/2 prefer to interact with the hypo-phosphorylated BIK1 (Fig. 2 and Supplementary Figs. 6–7, 9), while the flg22-induced BIK1 phosphorylation is a prerequisite for its monoubiquitination mediated by RHA3A/B[13]. RHA3A/B-mediated BIK1 monoubiquitination is required for the release of BIK1 from FLS2, and BIK1^C4KR fails to dissociate from FLS2 upon flg22 treatment, but RGLG1/2 still promote the protein accumulation of BIK1^C4KR (Supplementary Fig. 15). Nevertheless, the exact outcomes of RGLG1/2-mediated BIK1 ubiquitination await future investigation.

BIK1 is mainly present in a hypo-phosphorylated form in the resting state[6,7] PUB25/26 selectively ubiquitinate and target hypo-phosphorylated BIK1 for proteasomal degradation[14]. We show that RGLG1/2 preferentially interact with the hypo-phosphorylated BIK1. PUB25 interacts with RGLG2 and mediates its degradation. In turn, RGLG2 represses the ubiquitin ligase activity of PUB25 (Fig. 7). Therefore, the interplay of RGLG1/2 and PUB25 maintains BIK1 homeostasis to ensure that vigorous yet appropriate immune responses are induced upon pathogen attack (Fig. 7). Upon flg22 perception by FLS2

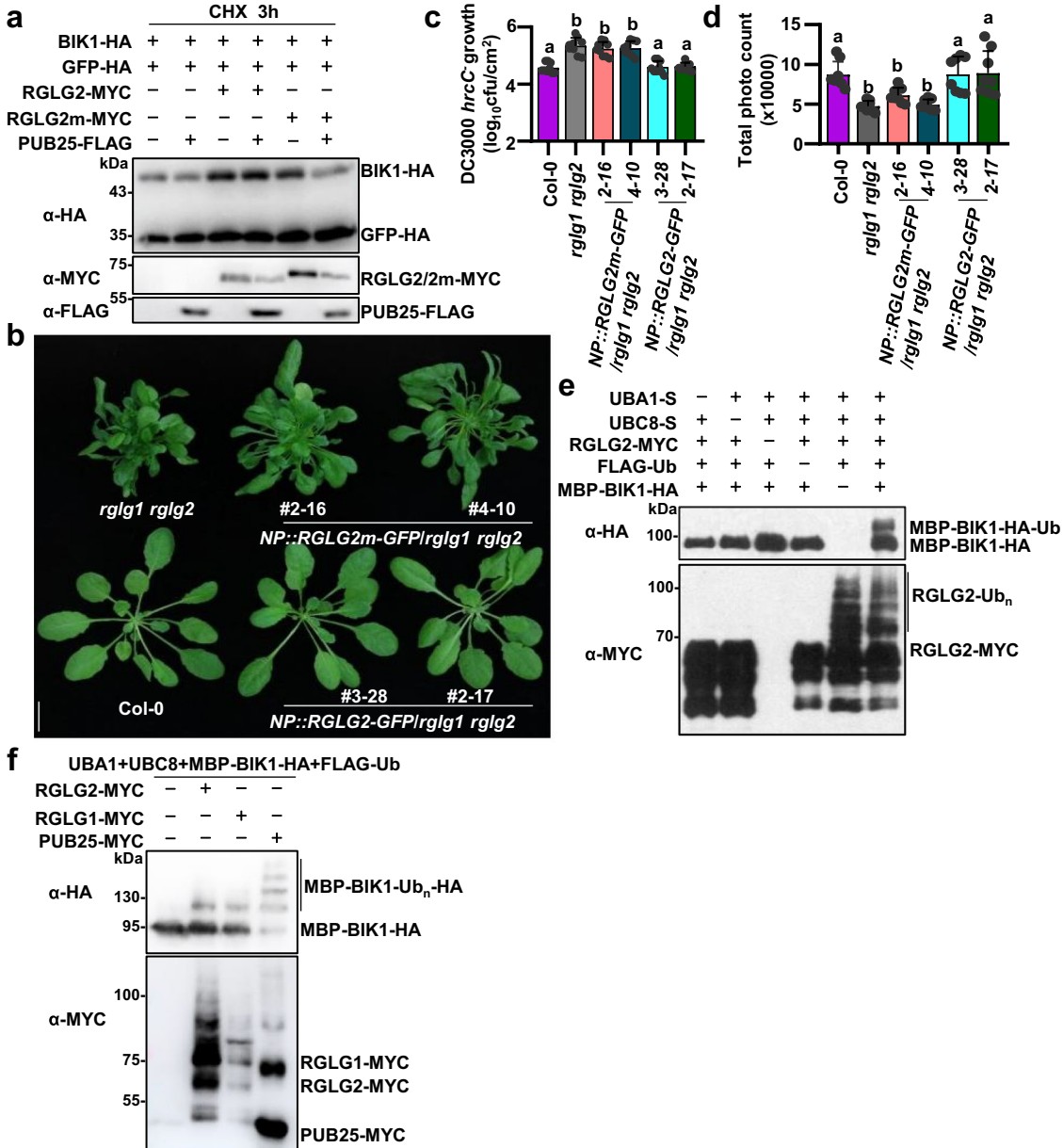

**Fig. 5 | The ubiquitin ligase activity is required for the role of RGLG2 in regulating immunity. a** RGLG2m does not suppress the PUB25-mediated BIK1 degradation. BIK1-HA and GFP-HA were co-expressed with RGLG2/RGLG2m-MYC and PUB25-FLAG in Arabidopsis protoplasts. The protoplasts were treated with 50 µM CHX for 3 h before harvesting. The transiently expressed proteins were detected by immunoblotting with the corresponding antibodies. Images shown are representative of three independent experiments. **b** *RGLG2m* partially rescues the growth phenotype of *rglg1 rglg2*. Four-week-old *pRGLG2(NP)::RGLG2/RGLG2m-GFP/rglg1 rglg2* transgenic lines were shown, scale bar: 1 cm. **c** Bacterial growth in *rglg1 rglg2* complementation lines. Leaves from 30-d-old *NP::RGLG2m-GFP/rglg1 rglg2* and *NP::RGLG2-GFP/rglg1 rglg2* plants were infiltrated with the *Pst* DC3000 *hrcC⁻*, and the bacterial growth was assessed 3 days post inoculation and was evaluated as colony-forming units per cm² of leaf area (CFU/cm²). Values are means ± SD (*n* = 9 leaves from three biological repeats using independent plants grown and inoculated under the same conditions. Different letters denote

significance difference (One-way ANOVA, $P < 0.0001$). Source data are provided as a Source Data file. **d** Flg22-induced ROS production in *rglg1 rglg2* complementation lines. Leaf discs of 5-week-old *NP::RGLG2m-GFP/rglg1 rglg2* or *NP::RGLG2-GFP/rglg1 rglg2* were treated with 200 nM flg22. ROS production was shown as total photo count (within 50 min of flg22 treatment). Values are means ± SD (*n* = 8 leave discs). Different letters denote significance difference (One-way ANOVA, $P < 0.0001$). Source data are provided as a Source Data file. **e** RGLG2 directly ubiquitinates BIK1. pACYC-RGLG2-MYC-AtUBC8-S, pCDF-MBP-BIK1-HA-AtUBA1-S, and pET-His-FLAG-Ub were co-expressed in *E. coli*. The bacterial lysates were subjected to immunoblotting analysis with anti-HA antibodies for detecting BIK1 ubiquitination or with α-MYC antibodies for detecting RGLG2 autoubiquitination. Images shown are representative of two independent experiments. **f** The ubiquitination pattern of BIK1 mediated by RGLG1/2 is different from that by PUB25. The bacterial ubiquitination assays were performed as described in (**e**). Images shown are representative of two independent experiments.

receptor complex, BIK1 is hyper-phosphorylated and the phosphorylation of PUB25/26 by CPK28 is also increased[14]. Furthermore, we showed that flg22 treatment induced the dissociation of RGLG1/2 and PUB25 (Fig. 4a, b). Altogether, these are thought to result in the enhanced ubiquitin ligase activity of PUB25/26 and depletion of

hypo-phosphorylated BIK1 to prevent overaccumulation of hyper-phosphorylated and activated BIK1, as previously proposed[14].

Interestingly, RGLG2 was shown to catalyze the formation of K63-linked ubiquitin chain[23]. RGLG1/2 are implicated in internalized protein trafficking between plasma membrane and endosomal compartments.

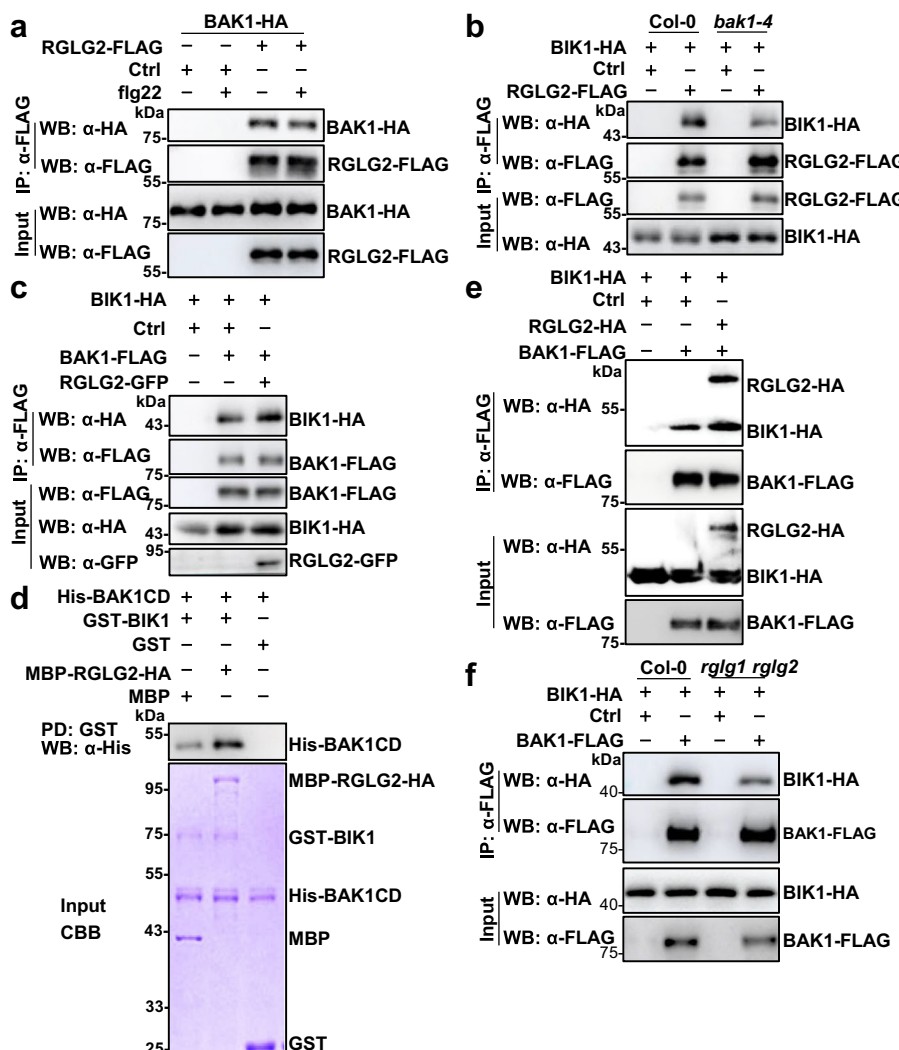

**Fig. 6 | RGLG2 promotes the association of BAK1 and BIK1. a** RGLG2 associates with BAK1. RGLG2-FLAG was co-expressed with BAK1-HA in Arabidopsis protoplasts, protoplasts were treated with or without 2.5 μM flg22 for 30 min, and IPs were performed using anti-FLAG antibodies. The associated BAK1-HA proteins were detected by immunoblotting with anti-HA antibodies. **b** RGLG2-BIK1 association partially requires BAK1. RGLG2-FLAG and BIK1-HA were co-expressed in Arabidopsis protoplasts isolated from *bak1* or Col-0 plants. IPs were performed using anti-FLAG antibodies. **a**, **b** Images shown are representative of two independent experiments. **c** RGLG2 promotes the association of BIK1 and BAK1. BIK1-HA and BAK1-FLAG were expressed together with or without RGLG2-GFP in Arabidopsis protoplasts. BAK1-FLAG proteins were immunoprecipitated with anti-FLAG antibodies. Images shown are representative of three independent experiments.

**d** RGLG2 promotes the interaction between BIK1 and BAK1 in vitro. The recombinant GST-BIK1 proteins immobilized on GSH beads were incubated with purified His-BAK1CD proteins in the presence or absence of MBP-RGLG2-HA. GST pull-down was performed and the pulled-down His-BAK1CD was detected by immunoblotting with anti-His antibodies. Input proteins were separated on SDS-PAGE and stained with CBB. **e** BAK1, BIK1, and RGLG1/2 are in the same complex. BAK1-FLAG was co-expressed with BIK1-HA and RGLG2-HA in protoplasts. BAK1-FLAG proteins were immunoprecipitated with anti-FLAG antibodies. **f** The association of BAK1 and BIK1 is weaker in *rglg1 rglg2* than in Col-0. BAK1-FLAG was co-expressed with BIK1-HA in *rglg1 rglg2* or Col-0 protoplasts. BAK1-FLAG proteins were immunoprecipitated with anti-FLAG antibodies. **d**–**f** Images shown are representative of at least two independent experiments.

The auxin transport protein PIN-FORMED 2 (PIN2) undergoes K63-linkeination mediated by RGLG1/2, resulting in the sorting of ubiquitinated PIN2 destined for proteolytic turnover[23,37]. RGLG2 also mediated the turnover of AtERF53, a positive transcriptional regulator controlling of drought responses[34]. Moreover, RGLG1 and RGLG5 were shown to catalyze the ubiquitination of PP2CA, a critical negative regulator of abscisic acid (ABA) signaling, resulting in its degradation by the 26S proteasome[35,38]. Following the submission of this work, Liu et al reported that the rice RGLG5 homolog, OsRGLG5 positively regulates the basal defense against *Magnaporthe oryzae* that causes rice blast. Furthermore, the *M. oryzae* effector AvrPi9 is ubiquitinated and degraded by OsRGLG5, and OsRGLG5 in turn is targeted by AvrPi9, which affects OsRGLG5 stability[39]. Taken together, RGLG1/2/5 have multiple substrates and are able to mediate the ubiquitination of these

substrates with different modes of ubiquitination, thereby being implicated in regulating diverse biological processes.

## Methods

### Plant materials

The Columbia-0 (Col-0) accession was used as wild-type (WT) *Arabidopsis thaliana*. The T-DNA insertion lines *rglg1-3/SALK_011892C*, *rglg2-2/SALK_062384C*[23], *pub25/SALK_147032*[14], *pub26/CS879195* were obtained from the Arabidopsis Biological Resource Center (ABRC), genotyped, and confirmed by RT-PCR analysis. The *rglg1 rglg2* double mutant was generated through a genetic cross between *rglg1* and *rglg2*, the *pub25 pub26* double mutant was generated through a genetic cross between *pub25* and *pub26*, and the double mutants were confirmed by genotyping and RT-PCR analysis. The primers used for genotyping are

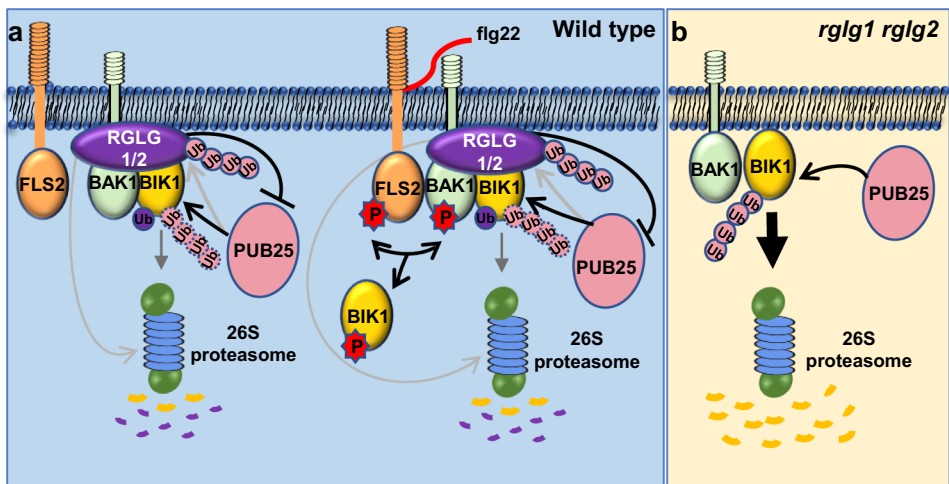

**Fig. 7 | Maintenance of BIK1 homeostasis by interplay between RGLG1/2 and PUB25/26 in Arabidopsis. a** RGLG1/2 suppress PUB25-mediated BIK1 degradation. BIK1 functions as an immune signaling hub and is associated with the FLS2-BAK1 receptor complex. BIK1 is mainly present in a hypo-phosphorylated form in the resting state; upon flg22 perception, BIK1 is hyper-phosphorylated. Two ubiquitin ligases PUB25/26 ubiquitinate the hypo-phosphorylated BIK1 and mediate its degradation. We proposed a model for maintenance of BIK1 homeostasis by interplay of different ubiquitin ligases. Two closely related RING-type ubiquitin ligases RGLG1 and RGLG2 specifically interact with the hypo-phosphorylated BIK1 and directly ubiquitinate BIK1. PUB25 mediates the degradation of RGLG2, and RGLG2 in turn suppresses the ubiquitin ligase of PUB25. RGLG1/2 suppress BIK1 degradation mediated by PUB25, promote BIK1 accumulation, and positively regulate immune signaling. **b** BIK1 protein accumulation is lower in *rglg1 rglg2* than in Col-0, and the immune responses are weaker in *rglg1 rglg2* than in Col-0.

listed in Supplementary Data 3. The *bak1-4* mutant has been reported previously[40].

## Constructs and generation of transgenic plants

For gene transient expression in protoplasts, the coding sequences (CDSs) of *RGLG1*, *RGLG2*, *RGLG3*, *PUB25*, *RHA3B*, *PBL13*, *ERF53*, *PP2CA*, *ATL80*, and *PUB39* were cloned into the *pHBT* vector, where they were fused to either an HA, FLAG, MYC, GFP, nYFP, or a cYFP tag. The PCR products were cloned into the *pHBT* vector to generate *pHBT-p35S::BIK1/FLS2/BAK1-HA/FLAG*[6,16], *pHBT-p35S::TaPBS1-RFP*[28], or *pHBT-p35S::BIK1-nYFP*[41]. For split-luciferase complementation assays, *BIK1*, *RGLG1*, and *RGLG2* were subcloned into the pCAMBIA1300-NLuc or pCAMBIA1300-CLuc vector; the CDSs of other ubiquitin ligases cloned into the pCAMBIA1300-HA-NLuc vector[42]. For constructs used for in vitro bacterial ubiquitination assays, the CDS of *RGLG1*, *RGLG2*, *RGLG2m*, *PUB25*, *BIK1*, or *PBL13* was subcloned in the Duet expression vector pCDFDuet™ or pACYCDuet™ (Novagen)[24]. *AtUBQ10* was subcloned into in the modified His-tagged fusion protein expression vector *pET28a* (Novagen) from a plant expression vector *pHBT-p35S::FLAG-UBQ10* (*Ub*) to generate *pET28a-His-FALG-Ub*[24,43]. For recombinant protein expression, CDSs of *RGLG1*, *RGLG2*, *RGLG3*, *PUB25*, and various truncated *RGLG2* were subcloned in the modified His-tagged, GST-tagged, or MBP-tagged fusion protein expression vector *pET28a* (Novagen), *pGEX-6P-1* (*pGEX4T-1*, Pharmacia), or *pMAL-C2* (New England Biolabs). *FLS2CD* and *BIK1* were subcloned in the modified *pGEX-6P-1* to generate *pGEX-FLS2CD* and *pGEX-BIK1*, respectively; *BAK1CD* was subcloned in the modified *pET28a* to generate *pET28a-BAK1CD* respectively[6,41,43].

For generation of *pBIK1::BIK1-HA* transgenic plants, *BIK1* CDS was cloned into the *pTF101* vector, where it was fused to an *HA* tag and under the control of its native promoter (1.9-kb upstream of its start codon, amplified by PCR from Col-0 genomic DNA). For generation of *pRGLG2::RGLG2-GFP* and *pRGLG2::RGLG2m-GFP* transgenic plants, *RGLG2* CDS or its mutant variant *RGLG2m* was cloned into the *pTF101* vector, where it was fused to a *GFP* tag and was under the control of its native promoter (-2.5-kb upstream of its start codon). *RGLG1/2-HA/BIK1-FLAG* were also subcloned into the the *pTF101* vector, where they were driven by the cauliflower mosaic virus 35S promoter. The plasmids were transformed into *Agrobacterium tumefaciens* GV3101, and then were transformed into Col-0 or *rglg1 rglg2*. Primers are listed in Supplementary Data 3. *pRGLG2::RGLG2-HA/p35S::BIK1-FLAG* plants and *p35S::RGLG1-HA/p35S::BIK1-FLAG* plants were generated through genetic crosses.

## Transcriptome sequencing

Twelve-day-old Col-0 seedlings were treated with 5 μM flg22 for 0, 15, 30, 60, 120, 240, 360, 480 min. Total RNA was extracted using TRIzol reagent (Invitrogen) following manufacturer's instructions. Library construction and transcriptome sequencing (three biological replicates) were performed by Novogene (Tianjin, China). Raw reads were processed through in-house Perl scripts. Q20, Q30, and GC content were calculated for the clean data. The clean reads were aligned to Arabidopsis genome using Hisat2 v2.0.5[44]. The read counts for each gene were generated using StringTie v1.3.3b[45]. The number of fragments per kilobase of transcript sequence per million base pairs sequenced (FPKM) was quantified to calculate the gene expression level. The differentially expressed genes (DEGs) were identified using DESeq2 R package with thresholds of |log2FoldChange| > 1 (FDR < 0.05)[46]. Gene Ontology (GO) enrichment was analyzed using the cluster Profiler R package, with GO terms with corrected *p*-value < 0.05 being considered significantly enriched. For analysis of dynamic gene expression pattern, the STEM (Short time-series Expression Miner Version 1.3.13) software was used to group the gene clusters following the manual[20].

## Reverse transcription-quantitative polymerase chain reactions (RT-qPCR)

Total RNA (1 μg) was treated with DNase I (Promega, Cat. # M6101), from which the first-strand cDNA was synthesized in 20 μL reactions using a reverse transcription kit (Promega, CAT # A277A). Real-time PCR was then performed on a Bio-Rad CFX-96 Real-Time PCR system using a SYBR Green qPCR kit (Promega, Cat. # A600A). Gene expression levels were normalized to that of *GAPC*, a stably expressed reference gene[47].

## Split-luciferase complementation assay

BIK1-CLuc with E3-NLuc or BIK1-NLuc with RGLG1/2-CLuc were co-infiltrated into 5- to 7-week-old *N. benthamiana* leaves via the *A.*

*tumefaciens*-mediated transformation method. Two days post-inoculation, the *N. benthamiana* leaves co-expressing CLuc-BIK1 and E3s-NLuc-HA were sprayed with luciferase substrate (Promega, Cat. # E152A). Then the leaves were kept in the dark for 5 min and the luminescence signal was captured with a CCD camera (Fusion FX7; Vilber, Marne-la-Vallée, France). The luciferase activity of leaves were quantified by Kuant 1.5 (Vilber, France).

### Transient gene expression in protoplasts

Arabidopsis mesophyll protoplasts were transfected with the indicated *pHBT* plasmids following the protocol[48]. Then the protoplasts were incubated in buffer containing 0.5 M mannitol (pH 5.7) at room temperature for 10 h. The protoplasts were treated with 50 µM CHX (Sigma, Cat. # 239763-M) for 2–3 h before harvesting them. For BiFC assays, protoplasts were transfected with BiFC vectors, and the fluorescence was detected using a Leica TCS SP8 confocal laser scanning microscope (Leica Microsystems, Wetzlar, Germany).

### Co-immunoprecipitation

For co-IP assays in protoplasts, 0.5 ml Arabidopsis protoplasts were transfected with the indicated *pHBT* plasmids. Then the protoplasts were incubated for 10 h and treated with or without 2.5 µM flg22 for 30 min before harvesting. The protoplasts were then lysed in 500 µl protein extraction buffer (10 mM HEPES [pH 7.5], 10% glycerol, 1 mM EDTA, 100 mM sodium chloride [NaCl], 0.5% Triton X-100, 1× complete protease inhibitors [Roche, Cat. # 04693159001]). The lysed protoplasts were vortexed vigorously for 1 min and then centrifuged at $12,470 \times g$ at 4 °C for 10 min. The supernatant was incubated with 1.5 µl anti-FLAG antibodies (sigma, Cat. # F1804) at 4 °C with gentle shaking for 2 h, then 5 µl of protein-G-agarose beads (Roche, Cat. # 11243233001) were added and the supernatant was incubated for another 2 h under the same condition. The samples were centrifugated at $12,470 \times g$ at 4 °C for 1 min. The protein-G-agarose beads were collected and washed three times with 1000 µl washing buffer (10 mM HEPES [pH 7.5]), 10% glycerol, 1 mM EDTA, 100 mM NaCl, and 0.1% Triton X-100) and once with 1000 µl 50 mM Tris hydrochloride (pH 7.5). After centrifugation at $12,470 \times g$ at 4 °C for 1 min, the beads were mixed with 20 µl of 1× protein sample buffer (60 mM Tris hydrochloride [pH 6.8], 10% glycerol, 2% sodium dodecyl sulfate [SDS], 5% 2-mercaptoethanol, 0.002% bromophenol blue). Then the protein-G-agarose beads were boiled at 95 °C for 10 min and centrifugated at $12,470 \times g$ for 30 s. The supernatant was collected and subjected to immunoblotting analysis with anti-HA-peroxidase (1:3000, Roche, Cat. # 12013819001) and anti-FLAG-peroxidase (1:3000, Sigma, Cat. # A8592).

For co-IP assays using transgenic plants, 12-d-old *pRGLG2 (NP)::RGLG2-HA/p35S::BIK1-FLAG* and *pRGLG2(NP)::RGLG2-HA*/Col-0 seedlings were treated with 5 µM flg22 for 30 min. Then the total proteins were extracted in 500 µl protein extraction buffer from 0.5 g seedlings. The samples were centrifuged at $12,470 \times g$ at 4 °C for 10 min. BIK1-FLAG proteins were immunoprecipitated with anti-FLAG peroxidase. Other procedures are similar with those for co-IP using protoplasts.

### Recombinant protein expression, purification and GST pull-down assay

The recombinant proteins were expressed in *Escherichia coli* BL21 cells at 25 °C for 10 h after addition of 0.5 mM isopropyl-β-D-thiogalactoside (IPTG) (sigma, Cat. # V900917). His-FLAG-PUB25/BAK1CD/RGLG1/RGLG2 proteins were purified using an Ni-NTA agarose purification kit (Qiagen) according to the manufacturer's instructions. MBP-RGLG2/BIK1/RGLG1/RGLG3-HA and GST-PUB25/BIK1/FLS2CD proteins were purified following the manufacturer's instructions.

GST-PUB25/BIK1/FLS2CD was used to pull down potential interacting proteins in vitro. GST-PUB25/BIK1/FLS2CD or GST proteins immobilized on GSH beads (Sigma, Cat. # G4510) were incubated in the pull-down buffer (20 mM Tris-HCl [pH 7.5], 150-mM NaCl, 3 mM EDTA, and 0.5% [v/v] Triton X-100) at 4 °C for 2 h. The GSH beads were collected by centrifugation at $12,470 \times g$ at 4 °C for 1 min and washed 5 times with the washing buffer (20 mM Tris-HCl [pH 7.5], 3 mM EDTA, 150-mM NaCl). The interacting proteins were detected by immunoblotting anti-His antibodies (1:3000, Sigma, Cat. # SAB1306082), anti-FLAG-peroxidase, or anti-HA-peroxidase.

### In vitro bacterial ubiquitination assay

The *pCDFDuet-MBP-BIK1/PBL13-HA-UBA1-S*, *pACYCDuet-RGLG1/2/RGLG2m-MYC-UBC8-S*, and *pET28a-FLAG-UBQ10* plasmids were co-transformed into *E. coli* BL21 (DE3) competent cells[24]. *E. coli* strain harboring the above expression vectors were cultured in 2 ml Luria-Bertani (LB) liquid medium with corresponding antibiotics at 37 °C and the expression of recombinant proteins was induced by IPTG when the medium absorbance at 600 nm reached 0.4–0.6. Then the bacteria were further cultured at 28 °C for 10–12 h and then kept at 4 °C overnight. Bacteria were collected from 100 µl medium by centrifugation at $12,000 \times g$ for 5 min, then the pellet was resuspended with 100 µl 1× protein sample buffer and boiled at 95 °C for 5 min. Then the crude protein extracts were subjected to immunoblotting analysis with anti-HA-peroxidase, anti-FLAG-peroxidase, anti-Myc-peroxidase (1:3000, Sigma, Cat. # A5598).

### ROS burst assay

Leaf discs with 4 mm in diameter were collected from 30-d-old Arabidopsis plants grown in soil. The leaf discs were soaked in sterile water overnight in a 96-well plate to avoid possible wounding responses caused during collection. Then water was replaced with the ROS measurement buffer (10 µg·mL$^{-1}$ horseradish peroxidase [Sigma, Cat. # V900503], 50 µM luminol [Sigma, Cat. # A8511], and 100-nM flg22). Luminescence was immediately measured using a GLOMAX 96 microplate luminometer (Promega). The values for ROS production were represented by Relative Light Units (RLU).

### MAPK activation assay

Seven-day-old Arabidopsis seedlings grown on 1/2MS plates were transferred to sterile water in a 12-well plate and were kept overnight to avoid possible wounding responses caused during collection. The plants were treated with 100 nM flg22 for the indicated times. Total proteins of each sample were extracted from 12 seedlings in 80 µL of extraction buffer (50 mM Tris-HCl [pH 7.5], 5 mM EDTA, 150 mM NaCl, 1% [v/v] Triton X-100, 1 mM NaF, 1 mM Na$_3$VO$_4$, 1 mM DTT, 1× complete protease inhibitors [Roche]). The supernatant was collected by centrifugation at $13,000 \times g$ for 5 min at 4 °C, mixed with 2× protein sample buffer, and subjected to immunoblotting analysis. MPK6/3/4 activation was detected with anti-pErk1/2 antibodies (1:2000, Cell Signaling Technology, Cat. # 9101). GAPDH was used as the loading control, which was detected by immunoblotting with anti-GAPDH antibodies (1:3000, Proteintech, USA, Cat. # 60004-1).

### Pathogen infection assay

*Pseudomonas syringae* pv. *tomato* (*Pst*) strain DC3000 or DC3000 *hrcC⁻* was cultured at 28 °C in liquid King's B (KB) medium (supplemented with 50 µg/mL rifampicin). After collection by centrifugation at $1500 \times g$ for 10 min, the bacteria were washed with 10 mM MgCl$_2$, and then were diluted with 10 mM MgCl$_2$ to the desired density (OD$_{600}$ = 0.001). Leaves of 30-d-old plants grown in soil were infiltrated with *Pst* DC3000 or *Pst* DC3000 *hrcC⁻* solution using a needleless syringe. Three days post-inoculation, leaf discs were collected and ground in 100 µL H$_2$O. Tenfold serial dilutions of the bacteria solution were prepared. Then bacteria were grown on KB plates (50 µg·mL$^{-1}$ rifampicin Hope Bio-Technogy, China, Cat. # HB8445]) at 28 °C for 3 days, and bacterial colony forming units (CFUs) were counted.

## Statistical analysis

The significance of the statistical was analyzed by one-way ANOVA and Student's *t*-tests (Supplementary Tables 1–2), *P*-values were generated by GraphPad Prism 8.0.

## Reporting summary

Further information on research design is available in the Nature Portfolio Reporting Summary linked to this article.

## Data availability

Source data are provided with this paper. The uncropped versions of all blots are provided with this paper. The RNA-seq data were deposited into National Center for Biotechnology Information and are accessible via BioProject accession number GSE227628. Source data are provided with this paper.

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

## Acknowledgements

We thank Drs Ping He and Libo Shan (Texas A&M University) for various *FLS2/BAK1/BIK1* constructs and *bak1-4* seeds, Dr Jie Zhang for pCAMBIA1300-NLuc and pCAMBIA1300-CLuc vectors. We thank Dr. Guozhong Huang for assistance in RNA-seq data analysis. The work was supported by grants from the National Natural Science Foundation of China (31371247, 32070555) and the State Key Laboratory of Plant Genomics.

## Author contributions

D.L., J.B., Y.Z., and J.S. designed this study. J.B., Y.Z., J.S., K.C., Y.H., and M.D. performed biochemistry experiments. J.B., Y.Z., J.S., R.W., Y.H., Y.Zo., and K.C. cloned genes and generated the constructs. J.B. and J.S. performed genetics and genetic transformation experiments. J.B. performed RNA-seq analysis and most phenotype analysis. B.J. and Y.Z. conducted pathogen assays. D.L. wrote the paper with the contribution from J.B. and Y.Z.

## Competing interests

The authors declare no competing interests.
