## [Peer Review File · Nature Communications]

BIK1 protein homeostasis is maintained by the interplay of different ubiquitin ligases in immune signalingREVIEWER COMMENTS

Reviewer #1 (Remarks to the Author):

In the manuscript entitled "BIK1 protein homeostasis is maintained by interplay of different ubiquitin ligases to regulate immune signaling" Bai et al., reported new positive regulators, RGLG1 and RGLG2, which positively regulate BIK1 protein level. Both RGLGs preferentially associate with the hypo-phosphorylated BIK1, compete with PUB25 for binding to BIK1, and suppress PUB25-mediated BIK1 degradation. They also found RGLGs directly ubiquitinate BIK1 and their ubiquitination activity is required for the suppression of PUB25-mediated BIK1 degradation. I think the paper is well written and the authors performed many different experiments to show the function of RGLGs on BIK1 homeostasis. Thus, I believe that the manuscript is potentially interesting for the readers in Nature com. However, I have a few concerns about some of their conclusions just based on the data in the manuscript.

Major points

The authors show the interaction of RGLGs to BIK1 only by the transient assay using Arabidopsis protoplast and the *N. benthamiana* system. The authors found RGLGs by similar expression profile to BIK1 but not by non-biased interaction screening such as Y2H and IP-masspec. Thus, the authors should be more cautious and try to see the interaction in more natural conditions by making stable transgenic lines expressing tagged proteins driven by their own promoters. The authors should also show the PM localization of RGLGs by experimental data.

rglg1 rglg2 mutant shows severe morphological phenotype, showing pleiotropic effects by the loss of RGLGs. In actuality, RGLG2 was shown to regulate PIN2 and AtERF53. The pleiotropic effects might explain the weak susceptible phenotype to Pst *hrcC*- and reduced ROS production. What do the *p35S::BIK1-FLAG/rglg1 rglg2* transgenic plants look like compared to *p35S::BIK1-FLAG/Col-0*? What is the kinetics of ROS production in *rglg1 rglg2*? BIK1 is not involved in *flg22*-inducible MAPKs activation. If RGLGs specifically regulate BIK1, *flg22*-inducible MAPKs activation is not changed in *rglg1 rglg2*. I think it is important to check these to show the specific regulation of BIK1 by RGLGs in PTI pathway.

In Figure 5c, the authors claimed that RGLG2 enhances BAK1-BIK1 interaction. The increased association is quite weak. Moreover, the slightly increased BIK1 protein amount in the output might be due to the increased BIK1 protein level in the presence of RGLG2 and not due to the enhanced interaction.

The authors show the importance of the ubiquitination activity of RGLGs for BIK1 regulation by using RGLG2m which carries mutations on 4 cysteine residues. In general, mutations in the cysteine residues can change the structure of the protein. Does RGLG2m interact with BIK1 similar to the WT RGLG2 *in vitro* and *in vivo*? It is possible that the mutations may change the structure of RGLG2 and reduce binding affinity to BIK1, which causes the loss of regulation of BIK1 and the ubiquitination activity may not be important for BIK1 regulation.

Other points

Line 105-106 and 109-110, please update this part with the most recent publication:
<https://doi.org/10.15252/embj.2020107257>

For figure 2b and supp figure 3a, it is strange to not express BIK1 as a negative control. The authors should use a closely related protein (for example PBL13) as an additional negative control.

For figure 3e, the BIK1-FLAG in *Col-0* is barely detectable, please include an image with stronger exposure.

There seems to be a weaker expression of RGLG1 and RGLG2 when co-expressed with

PUB25 (Figures 4a and 4b). Are there any direct interactions between RGLG1/2 and PUB25?

The label of figure 5a might be wrong. I believe it should be '- - + +' for RGLG2.

If both PUB25 and RGLG1/2 can ubiquitinate BIK1, how does one stabilize the protein and the other degrades it? What is the specificity? I think the authors should at least compare the BIK1 ubiquitination pattern induced by RGLG1/2, PUB25, and RHA3B.

In figure 6f, lane 5 of the western blot, why is there a band around 55KDa when PUB25 is not expressed? Is that a non-specific band?

Line 342, please keep the protein names (ALT44/45 and RHA3B) consistent throughout the paper.

Reviewer #2 (Remarks to the Author):

In this study, the authors have discovered that two RING-type ubiquitin ligases, RGLG1 and RGLG2, which are closely related, compete with PUB25 for associating with BIK1. Additionally, RGLG1/2 directly ubiquitinate BIK1, and their functions in favourably controlling immunological signals depend on the activity of the ubiquitin ligase. The study is novel and interesting and reveals how the ubiquitylation of BIK1 activates the immune response. Although the findings from the study are robust, there are some issues with the data, which I have listed below:

1. In this study, the RGLG1 was identified from the transcriptomic study; however, please explain why sometimes RGLG1 was used, and sometimes RGLG2 but claiming both to have a similar effect. It is important to maintain consistency in the results by including both RGLG1 and RGLG2 in all the experiments done in the study.
2. It is important to show in the supplementary data the different isoforms of RGLG and how much conservation they possess.
3. While performing the split luciferase complementation assays, why did the relative luciferase activity fluctuate in each setup, especially in Fig S2c when the ATL83 association was measured? Also, from the figure, the treatment of flg22 did not significantly reduce the association of ATL32 and ATL6-NL with BIK1-CL. Why was BIK1 was cloned preferentially with CLuc and not with NLuc?
4. In Fig 2, it is important to include the loading control to show an equal amount of protein loaded in each well.
5. In Fig 2, the authors have claimed both RGLG1 and RGLG2 to show an association to BIK1; however, they have only shown the flg22-induced dissociation of RGLG2 and BIK1 through co-IP. Similar studies need to be undertaken with RGLG1. As mentioned in line 208, I could not find a validation wherein the BIK1 hyperphosphorylation leads to the inhibition of RGLG2 interaction. It would be important to show this in a phospho-mimic mutant of BIK1.
6. The BIK1 is a target for PBL25-mediated ubiquitination. In lines 244-246, the overexpression of BIK1 in the rglg1rglg2 background does not increase the plant immunity to DC3000 infection. So does BIK1 OE increase the levels of PBLs in the rglg double mutant background?
7. There appears to be a considerable size difference between RGLG1 and RGLG2. The BIK1-HA levels in Fig 3 g and h without RGLG1/2 proteins are dissimilar when they are just biological replicates. There should be consistency in the results.
8. In fig 4d, the input lane of the GST-BIK1 shows similar levels in lane1 and 2 when RGLG2 should show more levels of BIK1. Also similar results were seen in 4c, wherein the BIK1-FLAG levels were unaltered when present with PUB25 only (3rd lane)
9. BIK1 is plasma membrane-associated protein, RGLG1/2 is also plasma membrane-associated. BiFC or co-localization assays can be done to show the physical interaction on the cellular level.
10. A competition plot can be used to understand the competitive interaction between BIK1, PUB25 and RGLG1/2. An example of this can be found in this publication

(<https://www.sciencedirect.com/science/article/pii/S1046202301911788>)

11. BAK1 promotes the recruitment of RGL2 with BIK1 however in Fig 5a, we observe that flg22 treatment does not promote the interaction of BAK1 with RGLG2. We find RGLG in the input lane even when RGLG is absent, which could be a non-specific band.

12. RGLG2 mediated interaction of BIK1 and BAK1 shown in Fig 5C is unclear. Pull down with FLAG followed by probing with α -GFP needs to be shown to validate the interaction of the RGLG2-BIK1-BAK1 complex.

13. In fig 5f lane 5 we observe low levels of PUB25 even when it is absent from the interaction.

14. The discussion section is repetitive in some places, for example, in lines 345-353.

15. It is strange why in Fig 4d, the size of His-FLAG-PUB25 is around 43kDa when blotted with α FLAG however, in Input, the protein size is around 55kDa. The experiment needs to be redone.

16. The working model mentioned that ubiquitination of BIK1 prevents its proteasomal degradation. It would be good to find the downstream effector of RGLG1/2 mediated monoubiquitinated BIK1. How is the monoubiquitylation of BIK1 different from that mediated by RHA3A/B?

It is intriguing how RGLG1/2 plays a vital role in the mono-ubiquitination of BIK1 and maintains its levels. For the other role of RGLG1/2, they induce proteolytic turnover of PIN2, ERF53 and PP2CA. However, it is unclear how the ubiquitination of BIK1 helps to prevent its degradation and promote immune responses. Overall, the study deals with an exciting dimension of BIK1-mediated regulation of plant immunity and incorporating the changes will make the findings more robust and straightforward for a broad readership.

Reviewer #3 (Remarks to the Author):

The manuscript by Bai and Sun et al. showed that the closely related ubiquitin ligases RGLG1/2 associate with the hypo-phosphorylated BIK1 and are required for the accumulation BIK1 in Arabidopsis plants. RGLG1/2 also associate with the receptor-like kinase BAK1, which is partially required for their association with BIK1. In addition, RGLG1/2 were shown to promote ubiquitination of BIK1 in vitro and compete with PUB25 for the association with BIK1. While these findings are interesting, additional data are required to clarify the mechanism of how RGLG1/2 regulate the stability of BIK1.

1. It was proposed that RGLG1/2 increase BIK1 stability through direct ubiquitination of BIK1 (mainly based on in vitro evidence that RGLG1/2 ubiquitinate BIK1). If this is true, a considerable amount of ubiquitinated BIK1 protein should be observed on various western blots shown in the manuscript, especially when RGLG1/2 were co-expressed with BIK1-KA, as ubiquitinated and non-ubiquitinated BIK1 proteins can be easily separated by SDS-PAGE.

2. RGLG1/2 were shown to compete with PUB25 for interaction with BIK1. Are the ubiquitin ligase activities of RGLG1/2 required for their competition with PUB25 for BIK1 interaction?

3. Can RGLG1/2 ubiquitinate PUB25 to affect its ability to degrade BIK1?

4. The *rglg1/2* double mutant exhibits pleiotropic phenotype as shown in Fig. 6B. Is expression of defense marker genes such as PR1 and PR2 up-regulated in the *rglg1/2* double mutant like in the *bik1*? Does overexpression of BIK1 alleviate the morphologic phenotype of *rglg1/2*?

Reviewer #1:

In the manuscript entitled “BIK1 protein homeostasis is maintained by interplay of different ubiquitin ligases to regulate immune signaling” Bai et al., reported new positive regulators, RGLG1 and RGLG2, which positively regulate BIK1 protein level. Both RGLGs preferentially associate with the hypo-phosphorylated BIK1, compete with PUB25 for binding to BIK1, and suppress PUB25-mediated BIK1 degradation. They also found RGLGs directly ubiquitinate BIK1 and their ubiquitination activity is required for the suppression of PUB25-mediated BIK1 degradation. I think the paper is well written and the authors performed many different experiments to show the function of RGLGs on BIK1 homeostasis. Thus, I believe that the manuscript is potentially interesting for the readers in Nature com. However, I have a few concerns about some of their conclusions just based on the data in the manuscript.

We appreciate your great suggestions and comments!

Major points

1. The authors show the interaction of RGLGs to BIK1 only by the transient assay using Arabidopsis protoplast and the *N. benthamiana* system. The authors found RGLGs by similar expression profile to BIK1 but not by non-biased interaction screening such as Y2H and IP-masspec. Thus, the authors should be more cautious and try to see the interaction in more natural conditions by making stable transgenic lines expressing tagged proteins driven by their own promoters. The authors should also show the PM localization of RGLGs by experimental data.

Thank you so much for your valuable suggestions!

1) We have performed co-IP assays using stable transgenic lines. We confirmed that RGLG2 was associated with BIK1 in *pRGLG2::RGLG2-HA/p35S::BIK1-FLAG* transgenic plants (Fig. 2d), and RGLG1 was associated with BIK1 in *p35S::RGLG1-HA/p35S::BIK1-FLAG* transgenic plants (Supplementary Fig. 6e). Consistently, the association of RGLG1/2 with BIK1 was reduced upon flg22 treatment (Fig. 2d and Supplementary Fig. 6e).

2) We demonstrated the PM localization of RGLGs by experimental data:

We showed that both RGLG1-GFP and RGLG2-GFP were localized to the plasma membrane, when they were expressed in Arabidopsis protoplasts (Supplementary Fig. 4c, d).

3) We also performed bimolecular fluorescence complementation (BiFC) assays in Arabidopsis protoplasts using BIK1 tagged with an nYFP (the N-terminal half of yellow fluorescent protein) tag and RGLG1/2 tagged with a cYFP (the C-terminal half of YFP) tag. BIK1 associated with RGLG1/2 at the plasma membrane. Whereas, BIK1-nYFP did not associate with RGLG3-cYFP, and RGLG1/2-cYFP did not associate with PBL13-nYFP (Fig. 2e and Supplementary Fig. 6f).

2. *rglg1 rglg2* mutant shows severe morphological phenotype, showing pleiotropic effects by the loss of RGLGs. In actuality, RGLG2 was shown to regulate PIN2 and AtERF53. The pleiotropic effects might explain the weak susceptible phenotype to Pst hrcC- and reduced ROS production. What do the *p35S::BIK1-FLAG/rglg1 rglg2* transgenic plants look like compared to *p35S::BIK1-FLAG/Col-0*? What is the kinetics of ROS production in *rglg1 rglg2*? BIK1 is not involved in flg22-inducible MAPKs activation. If RGLGs specifically regulate BIK1, flg22-inducible MAPKs activation is not changed in *rglg1 rglg2*. I think it is important to check these to show the specific regulation of BIK1 by RGLGs in PTI pathway.

Thanks a lot for these insightful suggestions!

1) The growth defect phenotype of *rglg1 rglg2* was almost not restored by overexpressing *BIK1* (Supplementary Fig. 12b), suggesting that the growth defect of *rglg1/2* is not caused by the reduced accumulation of BIK1 proteins.

2) We provided the kinetics of ROS production in *rglg1 rglg2* (New Fig. 3a).

3) We examined flg22-inducible MAPKs activation, and found that it was indeed not changed in *rglg1 rglg2* (Fig. 3b).

These results showed the specific regulation of BIK1 by RGLGs in PTI pathway.

3. In Figure 5c, the authors claimed that RGLG2 enhances BAK1-BIK1 interaction. The increased association is quite weak. Moreover, the slightly increased BIK1

protein amount in the output might be due to the increased BIK1 protein level in the presence of RGLG2 and not due to the enhanced interaction.

Thanks a lot for this good question!

We performed more experiments to further support that RGLG2 enhances BAK1-BIK1 interaction.

1) We expressed BAK1-FLAG together with BIK1-HA and RGLG1/2-HA in protoplasts and performed co-IP assays. The results not only showed that both BIK1-HA and RGLG1/2-HA could be pulled down by BAK1-FLAG, but also confirmed that the association of BAK1 and BIK1 were enhanced by RGLG/2 (Fig. 6e, Supplementary Fig. 20c).

2) Furthermore, we examined the the association of BAK1 and BIK1 in *rglg1 rglg2* mutant. We found that the association of BAK1 and BIK1 was weaker in *rglg1 rglg2* than in Col-0 (Fig. 6f).

4. The authors show the importance of the ubiquitination activity of RGLGs for BIK1 regulation by using RGLG2m which carries mutations on 4 cysteine residues. In general, mutations in the cysteine residues can change the structure of the protein. Does RGLG2m interact with BIK1 similar to the WT RGLG2 *in vitro* and *in vivo*? It is possible that the mutations may change the structure of RGLG2 and reduce binding affinity to BIK1, which causes the loss of regulation of BIK1 and the ubiquitination activity may not be important for BIK1 regulation.

Thank you for this good question!

To exclude this possibility, we examined the association of BIK1 with RGLG1m or RGLG2m. The co-IP results showed that the association of BIK1 with RGLG1/2 was comparable to that with RGLG1m/2m *in vivo*; and the *in vitro* pull-down results showed that the direct interaction of BIK1 with RGLG1/2 was comparable to that with RGLG1m/2m (Supplementary Fig. 17b-e).

Other points

Line 105-106 and 109-110, please update this part with the most recent publication: <https://doi.org/10.15252/embj.2020107257>

Thank you for this suggestion! We have updated the related parts in both introduction and discussion parts.

“Interestingly, PUB4 was also shown to have a dual effect on BIK1 homeostasis: it mediates the ubiquitination and degradation of non-activated BIK1 at the resting state, but promotes the accumulation of activated BIK1 after PAMP treatment”.

For figure 2b and supp figure 3a, it is strange to not express BIK1 as a negative control. The authors should use a closely related protein (for example PBL13) as an additional negative control.

Thank you for this suggestion! We performed new experiments using PBL13 as a negative control. We found that neither RGLG1 nor RGLG2 was able to associate with PBL13 (Supplementary Fig. 6c, d).

For figure 3e, the BIK1-FLAG in Col-0 is barely detectable, please include an image with stronger exposure.

Thank you for this suggestion! We have added an image with stronger exposure (new Fig. 3f).

There seems to be a weaker expression of RGLG1 and RGLG2 when co-expressed with PUB25 (Figures 4a and 4b). Are there any direct interactions between RGLG1/2 and PUB25?

Thank you so much for this insightful question! The review #3 also raised the similar question.

Inspired by your great question, we explored the relationship between RGLG1/2 and PUB25, and found that there is interplay between RGLG1/2 and PUB25. These results support the consumption in the title of the paper “BIK1 protein homeostasis is maintained by interplay of different ubiquitin ligases to regulate immune signaling”

1) PUB25 directly interacts with RGLG1/2

The results of co-IP assays showed that RGLG1/2 associated with PUB25 in Arabidopsis protoplasts. Furthermore, the association was reduced upon flg22 stimulation (Fig. 4a, b). We also performed BiFC assays, the results showed that PUB25-nYFP associated with RGLG1/2-cYFP at the plasma membrane, while PUB25-nYFP did not associate with RGLG3-cYFP, and RGLG1/2-cYFP did not associate with PUB39-nYFP (Fig. 4c and Supplementary Fig. 15a). Moreover, the results of pull-down assays showed that RGLG1/2 directly interacted with PUB25 *in vitro* (Fig. 4d and Supplementary Fig. 15b). Moreover, we found that PUB25 directly ubiquitinates RGLG2m *in vitro* (Supplementary Fig. 15c).

2) PUB25 mediates the proteosomal degradation of RGLG1/2

Interestingly, we found that, in the presence of CHX, the protein levels of RGLG2 were reduced when they were co-expressed with PUB25, which was blocked by MG132, an inhibitor of 26S proteasome (Fig. 4e). Furthermore, when transiently expressed, RGLG1/2 protein levels were higher in *pub25 pub26* double mutant protoplasts than in Col-0 protoplasts in the presence of CHX (Fig. 4f, Supplementary Fig. 16a, b). These results suggest that PUB25 mediates the proteosomal degradation of RGLG1/2.

3) RGLG2 suppresses the ubiquitin ligase activity of PUB25

PUB25 has ubiquitin ligase activity as demonstrated by its auto-ubiquitination (Fig. 4g). Interestingly, expression of RGLG2, but not RGLG2m led to the reduction in PUB25 auto-ubiquitination (Fig. 4g). However, the non-covalent association of PUB25 with a Ub moiety was not affected (Fig. 4g). These results suggest that RGLG2 suppresses the ubiquitin ligase activity of PUB25.

The label of figure 5a might be wrong. I believe it should be '- - + +' for RGLG2.

Thank you so much for catching this error! We have corrected the wrong labeling (New Fig. 6a)

If both PUB25 and RGLG1/2 can ubiquitinate BIK1, how does one stabilize the

protein and the other degrades it? What is the specificity? I think the authors should at least compare the BIK1 ubiquitination pattern induced by RGLG1/2, PUB25, and RHA3B.

Thank you for this good suggestion! We compared the BIK1 ubiquitination pattern induced by RGLG1/2 and PUB25. We found that the ubiquitination pattern mediated by RGLG1/2 was different from that by PUB25. The ubiquitination of BIK1 by PUB25 exhibited a laddering pattern, therefore, the type of BIK1 ubiquitination by PUB25/26 seems to be polyubiquitination; while that by RGLG1/2 seemed to be monoubiquitination (Fig. 5f).

For RHA3A/B-mediated BIK1 monoubiquitination, we introduced combined mutations of the C-terminal four lysines (ubiquitination sites mediated by RHA3A/B) into BIK1 and generated BIK1^{C4KR}. BIK1^{C4KR} was not able to dissociate from FLS2 upon flg22 treatment (Supplementary Fig. 14a). However, the protein accumulation of BIK1^{C4KR} promoted by RGLG1/2 was comparable to that of WT BIK1; and PUB25 mediates the degradation of BIK1^{C4KR} as it does for WT BIK1 (Supplementary Fig. 14b, c).

We also added new discussions: “however, the context, mechanism, and outcome of BIK1 monoubiquitination mediated by RHA3A/B are distinct from those by RGLG1/2. RGLG1/2 prefer to target the hypo-phosphorylated BIK1 (Fig. 2 and supplementary Fig. 5, 6, 8), while the flg22-induced BIK1 phosphorylation is a prerequisite for its monoubiquitination mediated by RHA3A/B. RHA3A/B-mediated BIK1 monoubiquitination is required for the release of BIK1 from FLS2, and BIK1^{C4KR} fails to dissociate from FLS2 upon flg22 treatment, while RGLG1/2 still promote the protein accumulation of BIK1^{C4KR} (supplementary Fig. 14). Nevertheless, the exact outcomes of RGLG1/2-mediated BIK1 ubiquitination await future investigation”.

In figure 6f, lane 5 of the western blot, why is there a band around 55KDa when PUB25 is not expressed? Is that a non-specific band?

Thank you for pointing this out! We have performed new experiments (New Fig. 5a).

Line 342, please keep the protein names (ALT44/45 and RHA3B) consistent throughout the paper.

Thank you for pointing this out! We have changed the proteins names and kept the name of RHA3A/B consistent throughout the paper.

Reviewer #2 (Remarks to the Author):

In this study, the authors have discovered that two RING-type ubiquitin ligases, RGLG1 and RGLG2, which are closely related, compete with PUB25 for associating with BIK1. Additionally, RGLG1/2 directly ubiquitinate BIK1, and their functions in favourably controlling immunological signals depend on the activity of the ubiquitin ligase. The study is novel and interesting and reveals how the ubiquitylation of BIK1 activates the immune response. Although the findings from the study are robust, there are some issues with the data, which I have listed below:

We appreciate your great suggestions and comments!

1. In this study, the RGLG1 was identified from the transcriptomic study; however, please explain why sometimes RGLG1 was used, and sometimes RGLG2 but claiming both to have a similar effect. It is important to maintain consistency in the results by including both RGLG1 and RGLG2 in all the experiments done in the study.

Thank you for this good suggestion! Actually, RGLG1 was much harder to be handled than RGLG2 due to unknown reasons, especially for purification of RGLG1 DNA by maxi prep to do protoplast transfection, and for obtaining RGLG1 transformants in *Escherichia coli* to analyze ubiquitination. We tried our best and performed the

following new RGLG1 related experiments during revision. Eventually, the most data actually include both RGLG1 and RGLG2. We found that RGLG1 and RGLG2 function similarly in regulating immunity.

- 1) BiFC assays for RGLG1 and BIK1,
- 2) BiFC assays for RGLG1-PUB25,
- 3) co-IP assays for RGLG1-BIK1,
- 4) co-IP assays for RGLG1-BIK1^{K105E},
- 5) co-IP assays for RGLG1-BIK1^{2D},
- 6) co-IP assays for RGLG1m-BIK1,
- 7) co-IP assays for RGLG1-PBL13,
- 8) co-IP assays for RGLG1-BIK1 association in *p35S::RGLG1-HA/p35S::BIK1-FLAG* transgenic plants,
- 9) co-IP assays for the effect of RGLG1/1m expression on BIK1-PUB25 association,
- 10) co-IP assays to verify that BAK1, BIK1, and RGLG1 are in the same complex and RGLG1 enhances BAK1-BIK1 association,
- 11) Pull-down assays for RGLG1-PUB25,
- 12) Pull-down assays for RGLG1m-BIK1,
- 13) *In vitro* ubiquitination assays to examine that RGLG1 directly ubiquitinates BIK1 but not PBL13,
- 14) RGLG1 promotes the protein accumulation of BIK1^{C4KR},
- 15) RGLG1 protein levels are higher in *pub25 pub26* double mutant protoplasts than in Col-0 protoplasts in the presence of CHX,
- 16) RGLG1 mediates the degradation of ERF53 and PP2CA,
- 17) The effect of RGLG1m on the PUB25-mediated BIK1 protein degradation,

2. It is important to show in the supplementary data the different isoforms of RGLG and how much conservation they possess.

Thanks a lot for this good suggestion! We have showed the different isoforms of RGLG and how much conservation they possess in Supplementary Fig. 4a, b.

a, Amino acid sequence alignment of Arabidopsis RGLG1-RGLG5. Identical and

similar amino acid residues were displayed on black and gray backgrounds, respectively. Four conserved Cys sites that chelate Zn²⁺ in the RING domain were highlighted in red boxes.

b, Phylogenetic analysis of RGLGs. A neighbor-joining phylogenetic tree was constructed based on the deduced amino acid sequences of RGLGs using MEGA5.0.3 software.

3. While performing the split luciferase complementation assays, why did the relative luciferase activity fluctuate in each setup, especially in Fig S2c when the ATL83 association was measured? Also, from the figure, the treatment of flg22 did not significantly reduce the association of ATL32 and ATL6-NL with BIK1-CL. Why was BIK1 was cloned preferentially with CLuc and not with NLuc?

Thank you for these great questions!

1) The fluctuation of relative luciferase activity in each setup could be due to the age of plants and the age of tobacco leaves we used for the assays. Split-luciferase complementation constructs were co-infiltrated into 5- to 7-week-old *N. benthamiana* leaves via the *A. tumefaciens*-mediated transformation method. We and other groups show that plant immunity varies with age (Zou et al., 2018; Hu et al., 2023). Therefore, the resistance of plant to *A. tumefaciens* and *A. tumefaciens* transformation efficiency may vary with the age of plants and the age of leaves used in the assays, due to the large number of split-luciferase complementation assays conducted in our work.

To confirm the results obtained via split luciferase complementation assays, we also performed co-immunoprecipitation (IP) assays in *Arabidopsis* protoplasts to examine the association of BIK1 with ATL80 or ATL6, and similar results were obtained (Supplementary Fig. 3a, b).

Zou, Y. M. et al. Transcriptional Regulation of the Immune Receptor FLS2 Controls the Ontogeny of Plant Innate Immunity. *Plant Cell* **30**, 2779–2794 (2018).

Hu L, Qi P, Peper A, Kong F, Yao Y, Yang L. Distinct function of SPL genes in age-related resistance in *Arabidopsis*. *PLoS Pathog.* 19(3):e1011218. (2023)

2) We have modified these statements:

“Furthermore, the association of BIK1 with RGLG1 or ATL83 was significantly reduced upon flg22 stimulation, and that with ATL32 or ATL6 was slightly reduced after flg22 treatment (Supplementary Fig. 2c)”.

3) We performed new split-luciferase complementation assays using BIK1-NLu, and similar results were obtained when BIK1-NLuc and RGLG1/2-CLuc were transiently expressed in *N. benthamiana* (Supplementary Fig. 5a and 5b).

4. In Fig 2, it is important to include the loading control to show an equal amount of protein loaded in each well.

Thank you for this good suggestion! We have included the loading control in the Figure. At the same time, the same amount of DNA was transfected into protoplasts in different transfections to ensure equal protein input.

5. In Fig 2, the authors have claimed both RGLG1 and RGLG2 to show an association to BIK1; however, they have only shown the flg22-induced dissociation of RGLG2 and BIK1 through co-IP. Similar studies need to be undertaken with RGLG1. As mentioned in line 208, I could not find a validation wherein the BIK1 hyperphosphorylation leads to the inhibition of RGLG2 interaction. It would be important to show this in a phospho-mimic mutant of BIK1.

Thank you for this great suggestion!

1) We performed new experiments with RGLG1:

A. To confirm the association of RGLG1 with BIK1, we co-expressed FLAG-epitope-tagged RGLG1 and HA-epitope-tagged BIK1 in Arabidopsis protoplasts. RGLG1-FLAG proteins were immunoprecipitated by anti-FLAG antibodies, and BIK1-HA was present in the RGLG1-FLAG immunoprecipitates, and the association between BIK1 and RGLG1 was reduced when the protoplasts were treated with flg22 (Fig. 2b);

B. Similarly, RGLG1-HA were also present in the BIK1-FLAG immunoprecipitates, and the association of BIK1 and RGLG1 was reduced when the protoplasts were treated with flg22 (Supplementary Fig. 5c);

C. We also performed co-IP assays using stable transgenic lines. We found that RGLG1 was associated with BIK1 in *p35S::RGLG1-HA/p35S::BIK1-FLAG* plants, and the association of BIK1 and RGLG1 was reduced when the protoplasts were treated with flg22 (Supplementary Fig. 6e);

D. We found that the association of RGLG1 with the BIK1 kinase dead mutant, BIK1^{K105E} (BIK1Km) was stronger than with wild-type (WT) BIK1. Moreover, the flg22-induced dissociation was not observed between RGLG1/2 and BIK1Km (Fig. 2g). By contrast, the association of RGLG1 with the BIK1 phospho-mimetic mutant BIK1S236D/T237D (BIK1^{2D}) was weaker than with WT BIK1 (Supplementary Fig 8b).

Additionally, we also performed BiFC assays in Arabidopsis protoplasts using BIK1-nYFP and RGLG1-cYFP. BIK1 associated with RGLG1 at the plasma membrane (Fig. 2e). Moreover, RGLG1 was pulled down by BIK1 fused to a GST tag, but not by GST or GST-FLS2CD (the intracellular region of FLS2) in *in vitro* assays (Fig. 2f)

2). To support that BIK1 hyperphosphorylation leads to the inhibition of RGLG1/2 interaction, we performed experiments to examine the association of RGLG1/2 with the BIK1 phospho-mimetic mutant BIK1S236D/T237D (BIK1^{2D}). We found that the association of RGLG1/2 with BIK1^{2D} was weaker than that with BIK1 (Supplementary Fig 8b, c).

6. The BIK1 is a target for PBL25-mediated ubiquitination. In lines 244-246, the overexpression of BIK1 in the *rglg1rglg2* background does not increase the plant immunity to DC3000 infection. So does BIK1 OE increase the levels of PBLs in the *rglg* double mutant background?

Thanks for this good question! We examined whether *BIK1* OE increase the levels of PUBs in the *rglg* double mutant background, and found that the transcript levels of

PUB25 in *p35S::BIK1-FLAG/rglg1 rglg2* were comparable to those in *rglg1 rglg2* (Supplementary Fig. 12d); and protein accumulation of exogenous *PUB25-HA* was not increased by *BIK1* overexpression (Supplementary Fig. 12e).

7. There appears to be a considerable size difference between RGLG1 and RGLG2. The *BIK1-HA* levels in Fig 3 g and h without RGLG1/2 proteins are dissimilar when they are just biological replicates. There should be consistency in the results.

Thanks a lot for this suggestion! We have redone this experiment (New Fig. 3g).

8. In fig 4d, the input lane of the GST-*BIK1* shows similar levels in lane1 and 2 when RGLG2 should show more levels of *BIK1*. Also similar results were seen in 4c, wherein the *BIK1-FLAG* levels were unaltered when present with *PUB25* only (3rd lane)

Thanks a lot for your good questions!

1) In Fig. 4d (new Supplementary Fig. 19b), the proteins were purified from *E. coli* first and then they were incubated *in vitro*, so there is no cellular machinery in this pull-down assays. I think we might miss some important information about this assay, so we added more detailed description in the Figure legend:

“RGLG1/2, but not RGLG3 reduce the interaction between *BIK1* and *PUB25 in vitro*. The recombinant proteins were expressed in *E. coli* and were affinity purified, GST-*BIK1* proteins immobilized on GST beads were incubated with purified MBP-RGLG1/2/3-HA and His-FLAG-*PUB25* proteins with gentle shaking”.

2) We and other groups (Wang et al. 2018) showed that the degradation of *BIK1* mediated by *PUB25* was evident in the presence of cycloheximide (CHX), a protein synthesis inhibitor (Fig. 3e, h, i; Wang et al. 2018), so was the promotion of *BIK1* accumulation by RGLG1/2 (Fig. 3e, g-i). So we added CHX treatment labelling in all the related Figures.

Wang, J. L. et al. A Regulatory Module Controlling Homeostasis of a Plant Immune Kinase. *Mol. Cell* **69**, 493–504 (2018)

9. BIK1 is plasma membrane-associated protein, RGLG1/2 is also plasma membrane-associated. BiFC or co-localization assays can be done to show the physical interaction on the cellular level.

Thank you for this great suggestion!

We performed BiFC bimolecular fluorescence complementation (BiFC) assays in Arabidopsis protoplasts using BIK1 tagged with an nYFP (the N-terminal half of yellow fluorescent protein) tag and RGLG1/2 tagged with a cYFP (the C-terminal half of YFP) tag. BIK1 associated with RGLG1/2 at the plasma membrane. Whereas, BIK1-nYFP did not associate with RGLG3-cYFP, and RGLG1/2-cYFP did not associate with PBL13-nYFP (Fig. 2e and Supplementary Fig. 6f). These results verify the specificity of the association between BIK1 and RGLG1/2 at the plasma membrane.

We also performed co-localization assays. RGLG1-GFP or RGLG2-GFP was co-expressed with TaPBS1-RFP (a known PM associated protein) in Arabidopsis protoplasts. Green and red fluorescent proteins were visualized via confocal microscopy. We found that both RGLG1-GFP and RGLG2-GFP were localized to the plasma membrane (PM), when they were expressed in Arabidopsis protoplasts (Supplementary Fig. 4c, d).

10. A competition plot can be used to understand the competitive interaction between BIK1, PUB25 and RGLG1/2. An example of this can be found in this publication (<https://www.sciencedirect.com/science/article/pii/S1046202301911788>)

Thank you for this great suggestion! A competition plot would be very useful for understanding the competitive interaction between BIK1, PUB25 and RGLG1/2.

However, our data of protein levels are based on images in different blots probed with different antibodies, and are very hard to be used for performing the competition plot analysis, which definitely awaits future investigation.

11. BAK1 promotes the recruitment of RGL2 with BIK1 however in Fig 5a, we observe that flg22 treatment does not promote the interaction of BAK1 with RGLG2. We find RGLG in the input lane even when RGLG is absent, which could be a non-specific band.

Thanks a lot for this question! Yes, flg22 treatment does not promote the interaction of BAK1 with RGLG2. Together with other results, we think “that RGLG2 is likely accompanying BAK1 in the same complex to regulate immunity” (Supplementary Fig. 20a, b).

Thank you so much for catching this error! We have corrected the labeling error (New Fig. 6a).

12. RGLG2 mediated interaction of BIK1 and BAK1 shown in Fig 5C is unclear. Pull down with FLAG followed by probing with α -GFP needs to be shown to validate the interaction of the RGLG2-BIK1-BAK1 complex.

Thanks a lot for this great suggestion! We followed your suggestion and performed new co-IP experiments. When BAK1-FLAG was co-expressed with BIK1-HA and RGLG1/2-HA in protoplasts, the results of co-IP not only showed that both BIK1-HA and RGLG1/2-HA could be pulled down by BAK1-FLAG, but also confirmed that the association of BAK1 and BIK1 were promoted by RGLG1/2, and also suggest that that BAK1, BIK1, and RGLG1/2 are in the same complex (Fig. 6e, Supplementary Fig. 20c).

Furthermore, we compared the the association of BAK1 and BIK1 in Col-0 and *rglg1 rglg2* plants, we found that the association of BAK1 and BIK1 was weaker in *rglg1 rglg2* than in Col-0 (Fig. 6f).

13. In fig 5f lane 5 we observe low levels of PUB25 even when it is absent from the

interaction.

Thank you for pointing this out! We have performed new experiments (New Fig. 5a).

14. The discussion section is repetitive in some places, for example, in lines 345-353.

Thanks for this suggestion! We have rewrote this part.

“Here, we demonstrate that another two closely related RING-type ubiquitin ligases, RGLG1 and RGLG2 promote BIK1 accumulation and positively regulate BIK1-mediated immune signaling. Importantly, our work reveal that BIK1 protein homeostasis is maintained by interplay of RGLG1/2 and PUB25 (Fig. 7).”

15. It is strange why in Fig 4d, the size of His-FLAG-PUB25 is around 43kDa when blotted with α -FLAG however, in Input, the protein size is around 55kDa. The experiment needs to be redone.

Thank you so much for catching this error! The protein size of His-FLAG-PUB25 should be around 55kDa as shown in the new supplementary Fig. 19b, where both RGLG1 and RGLG2 were included in the same assay.

16. The working model mentioned that ubiquitination of BIK1 prevents its proteasomal degradation. It would be good to find the downstream effector of RGLG1/2 mediated monoubiquitinated BIK1. How is the monoubiquitylation of BIK1 different from that mediated by RHA3A/B? It is intriguing how RGLG1/2 plays a vital role in the mono-ubiquitination of BIK1 and maintains its levels. For the other role of RGLG1/2, they induce proteolytic turnover of PIN2, ERF53 and PP2CA. However, it is unclear how the ubiquitination of BIK1 helps to prevent its degradation and promote immune responses. Overall, the study deals with an exciting dimension of BIK1-mediated regulation of plant immunity and incorporating the changes will make the findings more robust and straightforward for a broad readership.

Thank you for these insightful questions and suggestions!

1) We generated combined mutations of the C-terminal four lysines (BIK1^{C4KR}) that are ubiquitination sites mediated by RHA3A/B. BIK1^{C4KR} was unable to dissociate

from FLS2 upon flg22 treatment (Supplementary Fig. 14a). However, the protein accumulation of BIK1^{C4KR} promoted by RGLG1/2 was comparable to that of WT BIK1; and PUB25 mediates the degradation of BIK1^{C4KR} as it does for WT BIK1 (Supplementary Fig. 14b, c). These results suggest that the role of RGLG1/2 in regulating BIK1 is distinct from that of RHA3A/B.

2) We also added new discussions in the discussion section:

“The type of BIK1 ubiquitination by PUB25/26 seems to be polyubiquitination¹⁴ (Fig. 5f), while RHA3A/B mediate the monoubiquitination of BIK1¹³. In this work, we demonstrate that RGLG1/2 directly ubiquitinate BIK1, and the mode of BIK1 ubiquitination by RGLG1/2 seems to be monoubiquitination as demonstrated by *in vitro* assays (Fig. 5e, f and supplementary Fig. 18). However, the context, mechanism, and outcome of BIK1 monoubiquitination mediated by RHA3A/B are distinct from those by RGLG1/2. RGLG1/2 prefer to target the hypo-phosphorylated BIK1 (Fig. 2 and supplementary Fig. 5, 6, 8), while the flg22-induced BIK1 phosphorylation is a prerequisite for its monoubiquitination mediated by RHA3A/B¹². RHA3A/B-mediated BIK1 monoubiquitination is required for the release of BIK1 from FLS2, and BIK1^{C4KR} fails to dissociate from FLS2 upon flg22 treatment, while RGLG1/2 still promote the protein accumulation of BIK1^{C4KR} (supplementary Fig. 14). Nevertheless, the exact outcomes of RGLG1/2-mediated BIK1 ubiquitination await future investigation”.

3) Consistently, in our system, we found that the expression of RGLG1/2 reduced the protein abundance of ERF53 and PP2CA, both were shown to be the substrates of RGLG1/2 (Supplementary Fig. 11b, c), suggesting that RGLG1/2 have multiple substrates and are able to mediate the ubiquitination of these substrates with different modes of ubiquitination, thereby being implicated in regulating diverse biological processes.

Reviewer #3 (Remarks to the Author):

The manuscript by Bai and Sun et al. showed that the closely related ubiquitin ligases RGLG1/2 associate with the hypo-phosphorylated BIK1 and are required for the accumulation BIK1 in Arabidopsis plants. RGLG1/2 also associate with the receptor-like kinase BAK1, which is partially required for their association with BIK1.

In addition, RGLG1/2 were shown to promote ubiquitination of BIK1 in vitro and compete with PUB25 for the association with BIK1. While these findings are interesting, additional data are required to clarify the mechanism of how RGLG1/2 regulate the stability of BIK1.

We appreciate your great suggestions and comments!

1. It was proposed that RGLG1/2 increase BIK1 stability through direct ubiquitination of BIK1 (mainly based on in vitro evidence that RGLG1/2 ubiquitinate BIK1). If this is true, a considerable amount of ubiquitinated BIK1 protein should be observed on various western blots shown in the manuscript, especially when RGLG1/2 were co-expressed with BIK1-KA, as ubiquitinated and non-ubiquitinated BIK1 proteins can be easily separated by SDS-PAGE.

Thanks a lot for this good question!

It is hard to see the ubiquitinated BIK1 protein on various western blots shown in the manuscript, possibly because the ubiquitinated BIK1 proteins only represent a small fraction of the total proteins in cell. Actually, similar phenomena were observed in other cases, such as co-expression of BIK1 with PUB25/26 in the presence of MG132, and co-expression of BRI1 with PUB13 in the presence of MG132 (Wang et al., 2018; Zhou et al., 2018).

Wang, J. L. et al. A Regulatory Module Controlling Homeostasis of a Plant Immune Kinase. *Mol. Cell* **69**, 493–504 (2018)

Zhou J, et al. Regulation of Arabidopsis brassinosteroid receptor BRI1 endocytosis and degradation by plant U-box PUB12/PUB13-mediated ubiquitination. *Proc Natl Acad Sci U S A*. **115**, E1906-E1915 (2018)

2. RGLG1/2 were shown to compete with PUB25 for interaction with BIK1. Are the ubiquitin ligase activities of RGLG1/2 required for their competition with PUB25 for BIK1 interaction?

Thank you for this insightful question! We performed new experiments to examine the effect of RGLG1m/2m on the interaction of PUB25 and BIK1. The co-IP results showed that association of BIK1 and PUB25 was also reduced by RGLG1m/2m expression, although to a less extent than by RGLG1/2 (Supplementary Fig. 19c, d),

suggesting that RGLG1m/2m are also able to compete with PUB25 for BIK1, and the ubiquitin ligase activity of RGLG1/2 is partially required for their competition with PUB25. However, RGLG2m only slightly but not significantly restored the immune responses of *rglg1 rglg2* with respect to resistance to *Pst* DC3000 *hrcC*⁻ (Fig. 5c). These results imply that the competition of RGLG1/2 with PUB25 likely make a minor contribution to the function of RGLG1/2 in regulating BIK1 homeostasis.

3. Can RGLG1/2 ubiquitinate PUB25 to affect its ability to degrade BIK1?

Thank you for this insightful question!

We performed new experiments and found that RGLG2 suppresses the ubiquitin ligase activity of PUB25, which could affect its ability to degrade BIK1. “PUB25 has ubiquitin ligase activity as demonstrated by its auto-ubiquitination (Fig. 4g). Interestingly, we found that expression of RGLG2, but not RGLG2m led to the reduction in PUB25 autoubiquitination (Fig. 4g). However, the non-covalent association of PUB25 with a Ub moiety was not affected (Fig. 4g)”.

We found that PUB25 directly interacts with RGLG1/2 at the plasma membrane (Fig. 4a-d, Supplementary Fig.15a, b).

Furthermore, PUB25 directly ubiquitinates RGLG2m *in vitro* (Supplementary Fig. 15c), and in the presence of CHX, the proteins levels of RGLG2 were reduced when they were co-expressed with PUB25, which was blocked by MG132, an inhibitor of 26S proteasome (Fig. 4e). Moreover, when transiently expressed, RGLG1/2 protein levels were higher in *pub25 pub26* double mutant protoplasts than in Col-0 protoplasts in the presence of CHX (Fig. 4f, Supplementary Fig. 16a, b). These results suggest that PUB25 mediates the proteosomal degradation of RGLG1/2.

4. The *rglg1/2* double mutant exhibits pleiotropic phenotype as shown in Fig. 6B. Is expression of defense marker genes such as PR1 and PR2 up-regulated in the *rglg1/2* double mutant like in the *bik1*? Does overexpression of BIK1 alleviate the morphologic phenotype of *rglg1/2*?

Thanks a lot for these questions!

We measured the transcript levels and found that unlike those in *bik1*, the *PRI* transcript levels were slightly but not significantly higher in *rglg1 rglg2* than in Col-0 (Supplementary Fig. 9d). The growth defect phenotype of *rglg1 rglg2* were almost not restored by overexpressing *BIK1* (Supplementary Fig. 12b), suggesting that the growth defect of *rglg1/2* is not caused by the reduced accumulation of BIK1 proteins.

REVIEWERS' COMMENTS

Reviewer #1 (Remarks to the Author):

In the revised manuscript, the authors provided many additional data which well supports their model of BIK1 regulation by RGLGs. I think the overall data is convincing and the revised paper is well-written. However, I want to raise a few very minor points which would help to improve this paper further before publication in Nature Communication.

In Figure 3, the authors showed that RGLG1/2 promotes BIK1 protein accumulation. However, in Fig6 RGLG2 expression does not affect the BIK1 protein level. This difference would be probably because of the presence or absence of CHX. If so, the authors should explain possible mechanisms of CHX on RGLG1/2-mediated BIK1 protein accumulation. Why does RGLG1/2-mediated BIK1 protein accumulation become obvious only in the presence of CHX, while BIK1 reduction in rglg1/2 is obvious without CHX?

In Figure 6, the authors showed that RGLG2 promotes the association between BIK1 and BAK1. Can the authors exclude the possibility that RGLG2 enhances the BIK1-BAK1 interaction through the accumulation of BIK1? Why BIK1 level is less in rglg1/2 mutant in Fig. 3F, but not in Fig. 6F? I think the authors should carefully explain these points to avoid confusing readers.

How conserved are RGLG1/2 in plants? Do they exist outside the Brassicaceae?

Since RGLG1/2 can also associate with BAK1, can authors check the native protein level of BAK1 in the rglg1/2 double mutant? (BAK1 expression level looks slightly weaker in rglg1/2 compared to WT in Figure 6f.)

Reviewer #2 (Remarks to the Author):

The authors have addressed most of my concerns and the manuscript is substantially improved. However, several points require further attention.

1. Regarding my previous comments wherein RGLG1 and RGLG2 were sometimes used, the authors mention using RGLG1 in most experiments, including co-ip and luciferase assay. I suggest showing RGLG1 and BIK1 interaction on the main fig instead of supplementary Fig 5c.

2. RGLG1 and RGLG2 bind to hypophosphorylated BIK1, to be shown in the model.

3. What is the expression pattern of the 179 ubiquitin ligase genes in the transcriptome study? Do they all have expression patterns? A supplementary table in this regard would be helpful.

4. As mentioned in my previous comments (No. 3). The association with BIK1 with ATL83 or ATL32 is inconsistent in the split luciferase assay. The authors commented that the difference in the ATL83 levels was due to the age of the plants used in the experiments. I would suggest all the experiments be done in plants of similar age. Further, in the co-ip experiments (Fig S3a,b) the ATL83/ATL32 levels show no significant difference upon flg22 treatment. I would suggest redoing the experiment.

5. It is evident from this study that flg22 causes BIK1 phosphorylation which reduces its

association with RGLG1. However, RGLG promotes immunity. During infection, how BIK1-RGLG1 interaction modulates immunity needs to be discussed. Could it be related to a PTI and ETI response?

Minor points:

1. The spelling of minutes in Fig 1b.
2. Fig S3a: The labelling of ATL80-HA

Reviewer #3 (Remarks to the Author):

My suggestions have been adequately addressed.

REVIEWERS' COMMENTS

Reviewer #1 (Remarks to the Author):

In the revised manuscript, the authors provided many additional data which well supports their model of BIK1 regulation by RGLGs. I think the overall data is convincing and the revised paper is well-written. However, I want to raise a few very minor points which would help to improve this paper further before publication in Nature Communication.

We highly appreciate your great suggestions and comments!

1. In Figure 3, the authors showed that RGLG1/2 promotes BIK1 protein accumulation. However, in Fig. 6 RGLG2 expression does not affect the BIK1 protein level. This difference would be probably because of the presence or absence of CHX. If so, the authors should explain possible mechanisms of CHX on RGLG1/2-mediated BIK1 protein accumulation.

Thank you so much for the valuable suggestions!

Yes, this difference would be probably due to the presence or absence of CHX. We have added the following new statements and new data:

1) “The protein synthesis inhibitor cycloheximide (CHX) is often used in the protein degradation assays to exclude the translational effect, so that the changes in protein levels should be the already translated proteins (Kong et al., 2015; Wang et al., 2018; Yu et al., 2022)”

2) We also added new data showing that “notably, in the absence of CHX, overexpression of RGLG1 hardly affected the BIK1 protein accumulation in protoplasts (Supplementary Fig. 12d). This might be because transient protein overexpression in Arabidopsis protoplasts results in synthesis of large amounts of BIK1 proteins. Therefore, the contribution of BIK1 degradation to its abundance becomes un conspicuous under this condition”.

2. Why does RGLG1/2-mediated BIK1 protein accumulation become obvious only in the presence of CHX, while BIK1 reduction in *rglg1/2* is obvious without CHX?

Thanks for this good question!

The reasons are similar with those of the question #1.

1) “The protein synthesis inhibitor cycloheximide (CHX) is often used in the protein degradation assays to exclude the translational effect, so that the changes in protein levels should be the already translated proteins (Kong et al., 2015; Wang et al., 2018; Yu et al., 2022)”. Therefore, “the protoplasts were treated with CHX for 2 h to exclude the translational effect. The results showed that both RGLG1 and RGLG2, but not RGLG3, promoted the accumulation of BIK1 proteins (Fig. 3g, Supplementary Fig. 14a)”.

2) the effect of knocking out *RGLG1/2* on BIK1 abundance in *rglg1/2* is a “life-long” process, while RGLG1/2 were transiently expressed in protoplasts to demonstrate “RGLG1/2 mediate BIK1 protein accumulation”. Together, these data show that RGLG1/2 are required for BIK1 protein accumulation.

3. In Figure 6, the authors showed that RGLG2 promotes the association between BIK1 and BAK1. Can the authors exclude the possibility that RGLG2 enhances the BIK1-BAK1 interaction through the accumulation of BIK1? Why BIK1 level is less in *rglg1/2* mutant in Fig. 3F, but not in Fig. 6F? I think the authors should carefully explain these points to avoid confusing readers.

Thanks a lot for this good question!

The reasons are similar with those of the questions #1 and #2.

1) When we performed co-IP assays, we didn’t add CHX. “notably, in the absence of CHX, overexpression of RGLG1 hardly affected the BIK1 protein accumulation in protoplasts (Supplementary Fig. 12d). This might be because transient protein overexpression in Arabidopsis protoplasts results in synthesis of large amounts of BIK1 proteins. Therefore, the contribution of BIK1 degradation to its abundance becomes un conspicuous under this condition”.

2) The effect of knocking out *RGLG1/2* on BIK1 abundance in *rglg1/2* is a “life-long” process (Fig. 3F), while in Fig. 6F, RGLG1/2 were transiently expressed in *rglg1/2* protoplasts in the absence of CHX, and “transient protein overexpression in Arabidopsis protoplasts results in synthesis of large amounts of BIK1 proteins,

therefore, the contribution of BIK1 degradation to its abundance becomes un conspicuous”.

4. How conserved are RGLG1/2 in plants? Do they exist outside the Brassicaceae?

Thanks a lot for this good question!

We have performed amino acid sequence alignment of RGLG1/2 homologs from Arabidopsis, rice (OsRGLG1/2), maize (ZmRGLG1/2), and *Brachypodium distachyon* (BdRGLG1/2) (new Supplementary Fig.5a). Both the C domain and the R domain are conserved in AtRGLGs and RGLG homologs, and the C425/C428/C454/C457 sites chelating Zn²⁺ in the RING domain of RGLG2 are conserved in AtRGLGs and RGLG1/2 homologs.

5. Since RGLG1/2 can also associate with BAK1, can authors check the native protein level of BAK1 in the *rglg1/2* double mutant? (BAK1 expression level looks slightly weaker in *rglg1/2* compared to WT in Figure 6f.)

Thanks a lot for this good question!

We checked the native protein level of BAK1 in the the *rglg1/2* double mutant using the anti-BAK1 antibodies, the results showed that BAK1 protein level was comparable in *rglg1 rglg2* and Col-0 (Supplementary Fig. 21a).

Reviewer #2 (Remarks to the Author):

The authors have addressed most of my concerns and the manuscript is substantially improved. However, several points require further attention.

We highly appreciate your great suggestions and comments!

1. Regarding my previous comments wherein RGLG1 and RGLG2 were sometimes used, the authors mention using RGLG1 in most experiments, including co-ip and luciferase assay. I suggest showing RGLG1 and BIK1 interaction on the main fig instead of supplementary Fig 5c.

Thanks for this good suggestion!

We have moved the RGLG1 and BIK1 interaction from supplementary Fig. 5c to the main Fig. 2d.

2. RGLG1 and RGLG2 bind to hypo-phosphorylated BIK1, to be shown in the model.

Thanks for this good suggestion!

We have modified the model.

Additionally, new statements were added in the legends, “a, RGLG1/2 suppress PUB25-mediated BIK1 degradation. BIK1 functions as an immune signaling hub and is associated with the FLS2-BAK1 receptor complex. BIK1 is mainly present in a hypo-phosphorylated form in the resting state; upon flg22 perception, BIK1 is hyper-phosphorylated. Two ubiquitin ligases PUB25/26 ubiquitinate the hypo-phosphorylated BIK1 and mediate its degradation”.

3. What is the expression pattern of the 179 ubiquitin ligase genes in the transcriptome study? Do they all have expression patterns? A supplementary table in this regard would be helpful.

Thanks a lot for this good suggestion, we have added the expression of the 179 ubiquitin ligase genes in the supplementary Dataset 2.

4. As mentioned in my previous comments (No. 3). The association with BIK1 with ATL83 or ATL32 is inconsistent in the split luciferase assay. The authors commented that the difference in the ATL83 levels was due to the age of the plants used in the experiments. I would suggest all the experiments be done in plants of similar age. Further, in the co-ip experiments (Fig S3a,b) the ATL83/ATL32 levels show no significant difference upon flg22 treatment. I would suggest redoing the experiment.

Thanks a lot for this great suggestion!

1) We have redone all the experiments in plants of similar age (7-week-old *N. benthamiana*). The results showed that: “of the 16 ubiquitin ligases tested, 6 ones (RGLG1, ATL83, ATL32, ATL6, ATL40, and ATL80) were found to be associated with BIK1 (Supplementary Fig. 2b). Furthermore, the association of BIK1 with RGLG1, ATL83, ATL32, or ATL6 was reduced upon flg22 stimulation as assayed by SLC in *N. benthamiana* (Supplementary Fig. 2c). We also confirmed the association of BIK1 with ATL6 by co-immunoprecipitation (IP) assays in Arabidopsis protoplasts. Notably, flg22 treatment reduced their association (Supplementary Fig. 3a). The association of BIK1 with ATL40 or ATL80 was almost not affected by flg22 treatment as assayed by SLC (Supplementary Fig. 2c), and that with ATL80 was also confirmed by co-IP assays in Arabidopsis protoplasts (Supplementary Fig. 3b)”.

2) Actually, Fig S3a-b showed the co-IP results of BIK1 with ATL6 or ATL80, not “ATL83/ATL32” in the previous submission. I am Sorry, I think we didn’t make it clear in the previous submission. We have redone the co-IP assay for confirming BIK1-ATL6 association, “we confirmed the association of BIK1 with ATL6 by co-immunoprecipitation (IP) assays in Arabidopsis protoplasts. Notably, flg22 treatment reduced their association (Supplementary Fig. 3a)”.

5. It is evident from this study that flg22 causes BIK1 phosphorylation which reduces its association with RGLG1. However, RGLG promotes immunity. During infection, how BIK1-RGLG1 interaction modules immunity needs to be discussed. Could it be related to a PTI and ETI response?

Thanks a lot for this insightful suggestion!

We have added a new paragraph in the Discussion section.

“BIK1 is mainly present in a hypo-phosphorylated form in the resting state (Lu et al., 2010; Zhang et al., 2010). PUB25/26 selectively ubiquitinate and target hypo-phosphorylated BIK1 for proteasomal degradation (Wang et al., 2018). We show that RGLG1/2 preferentially interact with the hypo-phosphorylated BIK1. PUB25 interacts with RGLG2 and mediates its degradation. In turn, RGLG2 represses the ubiquitin ligase activity of PUB25 (Fig. 7). Therefore, the interplay of RGLG1/2 and PUB25 maintains BIK1 homeostasis to ensure that vigorous yet appropriate immune responses are induced upon pathogen attack (Fig. 7). Upon flg22 perception by FLS2 receptor complex, BIK1 is hyper-phosphorylated and the phosphorylation of PUB25/26 by CPK28 is also increased (Wang et al., 2018). Furthermore, we showed that flg22 treatment induced the dissociation of RGLG1/2 and PUB25 (Fig. 4a, b). Altogether, these are thought to result in the enhanced ubiquitin ligase activity of PUB25/26 and depletion of hypo-phosphorylated BIK1 to prevent overaccumulation of hyper-phosphorylated and activated BIK1, as previously proposed (Wang et al., 2018)”.

Minor points:

1. The spelling of minutes in Fig 1b.

Thank you so much for pointing this error out! We have made correction.

3. Fig S3a: The labelling of ATL80-HA

Thanks for pointing this point out! We have made the change in the new Fig. S3b.

Reviewer #3 (Remarks to the Author):

My suggestions have been adequately addressed.

We highly appreciate your comments!

References

1. Kong, L. et al. Degradation of the ABA co-receptor ABI1 by PUB12/13 U-box E3 ligases. *Nat Commun.* **6**, 8630 (2015).
2. Wang, J. L. et al. A Regulatory Module Controlling Homeostasis of a Plant Immune Kinase. *Mol. Cell* **69**, 493–504 (2018).
3. Yu, G. et al. The Arabidopsis E3 ubiquitin ligase PUB4 regulates BIK1 and is targeted by a bacterial type-III effector. *EMBO J.* **41**, e107257 (2022).
4. Lu, D. P. et al. A receptor-like cytoplasmic kinase, BIK1, associates with a flagellin receptor complex to initiate plant innate immunity. *Proc. Natl. Acad. Sci. USA* **107**, 496–501 (2010).
5. Zhang, J. et al. Receptor-like cytoplasmic kinases integrate signaling from multiple plant immune receptors and are targeted by a *Pseudomonas syringae* effector. *Cell Host Microbe* **7**, 290–301 (2010).